# FedMH: Federated Learning with Multi-Task Head for Heterogeneous Models in Offline Signature Verification

## Abstract

Offline handwritten signature verification is a critical biometric authentication technology widely used in high-risk domains such as finance and law. However, data silos constrain models to train solely on local, limited, and imbalanced data, typically resulting in overfitting to majority classes. Federated learning is one of the solutions that addresses this issue by enabling collaborative training while preserving data privacy. However, existing federated learning methods face three key challenges in the scenario of offline handwritten signature verification: the scarcity of local data and class imbalance, the need for a dual-task head, and heterogeneity constraints in the model aggregation process. To tackle these issues, we propose FedMH, a novel federated learning method based on a multi-task head strategy, which supports heterogeneous model environments. Specifically, FedMH comprises three core components including (1) an adaptive data augmentation strategy that enhances data diversity and improves learning in minority classes by targeting few-shot classes in local single-user genuine and forged signature subsets, (2) a dual-task head collaboration mechanism that dynamically guides parameter updates to foster synergy between tasks, and (3) a gradient optimization method employing linear probing in parameter space and Pareto improvement criteria to enable efficient knowledge aggregation across heterogeneous models. We also provide a theoretical convergence proof for FedMH to ensure its reliability and stability. Comprehensive experiments validate the effectiveness of the proposed FedMH on three benchmark handwritten signature datasets. The proposed FedMH achieves state-of-the-art performance compared to heterogeneous federated baseline methods and solves the long-standing problem of multi-task head collaboration. At the same time, when faced with unfamiliar datasets, FedMH also demonstrates stronger generalization ability than the baseline methods.

## 1 Introduction

Offline handwritten signature verification is a critical biometric authentication technology that plays an essential security role in high-stakes domains such as finance and law (Hindumathi et al., 2023). However, the development of accurate and robust verification models has long been affected by the issue of data isolation, and the centralized collection of raw data is strictly prohibited. (Hafemann et al., 2019). This restriction forces models to train locally on limited datasets, compromising generalization against evolving forgery techniques. Additionally, hardware heterogeneity across clients further impedes effective collaborative training.

Federated learning has emerged as a promising distributed paradigm that enables collaborative training without sharing raw data, thereby offering a viable path to circumvent data isolation (Abdulrahman et al., 2021). However, a critical limitation is observed when applying existing federated learning methods directly to signature verification. Although standard federated learning approaches, and even some multi-task federated methods, have achieved success, they predominantly rely on a core assumption that each client focuses on a single learning task (Hu et al., 2022). This paradigm contradicts the practical requirements of offline handwritten signature verification, where each client must simultaneously handle two related yet competing tasks. Specifically, user verification is a

multi-class classification task, while forgery detection is a binary discrimination task to distinguish genuine from forged signatures (Zheng et al., 2025).

To address the above issues, a novel federated learning approach named FedMH is proposed. This method innovatively constructs a dual-task collaboration framework based on the concept of Pareto improvement. Additionally, FedMH effectively addresses the extreme scarcity of skilled forged samples and mitigates data imbalance issues for genuine and forged signatures within and across clients. Specifically, the proposed FedMH method formulates the dual-task head co-iteration problem within clients as a search for Pareto stationary solutions. Global communication is utilized to integrate knowledge from other heterogeneous clients, enhancing generalization ability. Effective knowledge sharing is achieved between the two tasks within each client and across different heterogeneous clients. The main contributions of this work are as follows.

- A client-adaptive minority class data augmentation module is proposed to balance and enhance local data. This module uses selective oversampling and data transformation techniques to address the class imbalance between genuine and forged signatures from individual users. It effectively reduces the bias of models toward majority classes and improves learning performance for minority classes.

- A novel dual-task head mechanism is employed for model updates on local clients. This mechanism dynamically aligns task-specific gradients, guiding the model toward a Pareto stationary solution that balances the performance of both tasks.

- An efficient Pareto gradient optimization protocol is employed for global updates in a heterogeneous environment of client models. Clients prioritize selecting gradient combinations that constitute Pareto-improving directions for the current model, thereby ensuring robust convergence and enhanced generalization.

## 2 RELATED WORK

### 2.1 HETEROGENEOUS FEDERATED LEARNING

In heterogeneous federated learning, clients not only have non-independent and identically distributed (non-IID) local data but also possess different model architectures, making direct parameter aggregation challenging (Ye et al., 2023). To address this structural disparity, FedKD achieves knowledge transfer between heterogeneous local models through logit-based distillation Chang et al. (2020); FedGH employs a hypernetwork to generate personalized parameters tailored to each client's model structure (Yi et al., 2023); FedProto aligns local representations by aggregating class prototypes without sharing model parameters (Tan et al., 2022); FML uses meta-learning to learn an architecture-agnostic initialization that can easily adapt to various local models (Shen et al., 2020); FedTGP reduces the transfer gap between heterogeneous models by aligning features and output distributions (Zhang et al., 2024); while FedGen (Zhu et al., 2021) and LG-FedAvg (Liang et al., 2020) promote indirect knowledge sharing through generative modeling or decoupled representation learning, respectively. These methods avoid direct parameter aggregation and instead leverage knowledge distillation, hypernetworks, prototype matching, or meta-learning to bridge architectural differences while maintaining personalization, collectively overcoming the challenge of model heterogeneity.

Although federated learning offers a promising paradigm for privacy-preserving collaboration, existing methods remain inadequate in the context of offline handwritten signature verification. This is because they focus on single-task head mechanisms and fail to address the extreme imbalance between genuine and forged handwritten signature samples.

### 2.2 HANDWRITTEN SIGNATURE VERIFICATION METHODS

Early offline handwritten signature verification methods relied on handcrafted descriptors such as Local Binary Patterns (LBP) and Histogram of Oriented Gradients (HOG), which lacked the representational capacity to handle complex forgeries (Yilmaz et al., 2011). Deep learning methods significantly advanced performance, demonstrating the effectiveness of representation learning in capturing fine-grained signature variations, SigNet employed convolutional neural networks (CNNs) to learn discriminative features (Hafemann et al., 2017); the Pseudo-Contrastive Features (PCF)

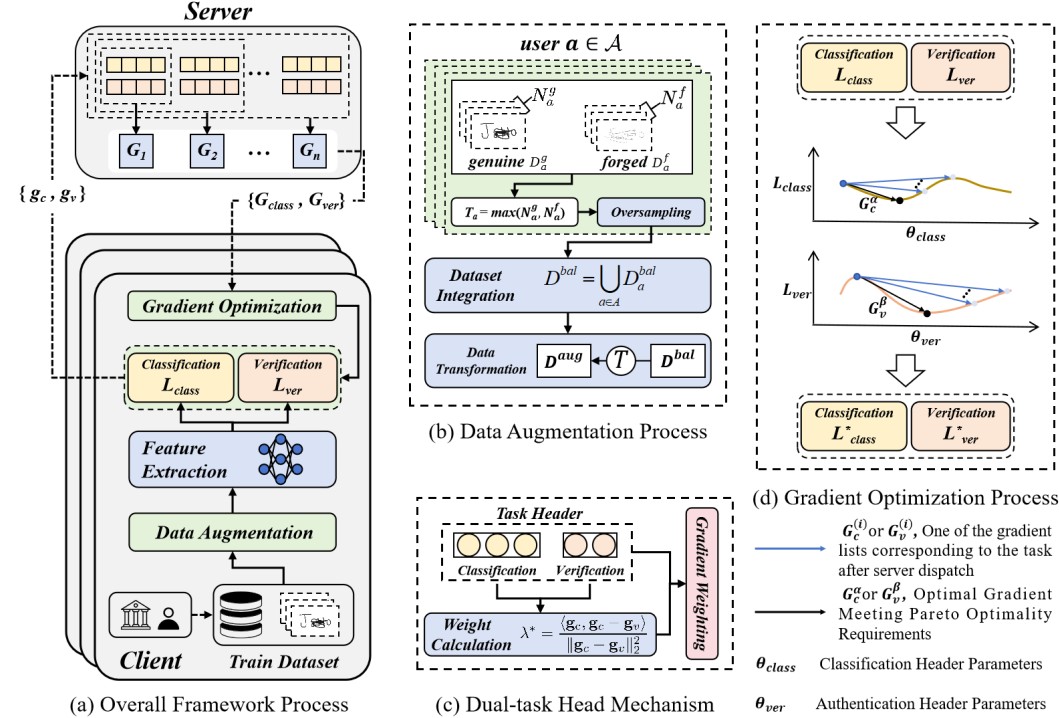

Figure 1: Architecture of the proposed FedMH framework. (a) Overall client-server communication and training workflow. (b) Client **data augmentation** module detail. (c) Client **dual-task head** mechanism detail. (d) Pareto **gradient optimization** module detail.

method enhanced the separability between genuine and forged signatures (Liu et al., 2021); rejection mechanisms and top-ranking strategies improved decision reliability (Maergner et al., 2019); while hybrid CNN-Transformer architectures further strengthened feature modeling (Li et al., 2024). Although these approaches established the value of deep representations, they relied on large-scale centralized data and assumed homogeneous data distribution—conditions rarely met in practical scenarios (Diaz et al., 2019). In few-shot and highly imbalanced settings, they tend to suffer from overfitting and task performance bias, while remaining incompatible with federated learning frameworks (Li et al., 2022; Maruyama et al., 2021). Consequently, centralized deep models alone are insufficient to meet the integrated requirements of handwritten signature verification regarding privacy preservation, multi-task collaboration, and data efficiency.

To address these limitations, this paper proposes FedMH, which formulates federated offline handwritten signature verification as an iterative process toward Pareto-stationary solutions for multi-task heads within clients. This approach establishes a generalizable paradigm for client-side dual-task head federated learning.

## 3 FEDERATED MULTI-TASK HEAD MECHANISM FOR OFFLINE HANDWRITTEN SIGNATURE VERIFICATION SCENARIOS

This section systematically presents the FedMH methodology. As illustrated in Figure 1, the framework comprises three novel components, including a client-adaptive data augmentation module for local data balancing, a dual-task head mechanism for local client updates, and an efficient Pareto gradient optimization protocol for heterogeneous client environments.

### 3.1 CLIENT-ADAPTIVE DATA AUGMENTATION

In federated offline handwritten signature verification, local client datasets often exhibit substantial disparity between the number of genuine signatures $N_a^g$ and forged signatures $N_a^f$ for individual

user $a$, resulting in model bias toward majority classes. To mitigate this bias, a client-adaptive data augmentation module is proposed that selectively enhances the minority class while preserving the semantic integrity of signature samples. Assume each client possesses a local dataset containing signatures from a user set $\mathcal{A}$. For each user $a \in \mathcal{A}$, the following datasets are defined,

$$\mathcal{D}_a^g = \{(x_i, a, 1) \mid i = 1, \ldots, N_a^g\}, \quad \mathcal{D}_a^f = \{(x_i, a, 0) \mid i = 1, \ldots, N_a^f\}, \tag{1}$$

where $x_i$ denotes the $i$-th signature image, $a$ is the user label, and the third component indicates the authenticity label (1 for genuine, 0 for forged), with $N_a^g$ and $N_a^f$ denoting the numbers of genuine and forged signatures for user $a$, respectively. It is commonly observed that $N_a^g \gg N_a^f$ due to the difficulty in collecting skilled forgeries.

This module balances the number of samples between the minority and majority class of genuine and forged signatures from user $a$. The target number of samples $T_a$ for each class is set as follows $T_a = \max(N_a^g, N_a^f)$. After determining the target number $T_a$ of genuine and forged signature samples for user $a$, an oversampled set $\mathcal{O}_a^g$ is constructed by uniformly random sampling with replacement from the genuine signature set $\mathcal{D}_a^g$ if $N_a^g < T_a$. Similarly, if $N_a^f < T_a$, an oversampled set $\mathcal{O}_a^f$ is generated from the forged signature set $\mathcal{D}_a^f$ following the same procedure. The oversampled sets can be uniformly expressed as,

$$\mathcal{O}_a^k = \{x_j \mid x_j \sim \text{Uniform}(\mathcal{D}_a^k), j = 1, \ldots, T_a - N_a^k\}, \tag{2}$$

where $k \in \{g, f\}$ corresponds to genuine signatures and forged signatures, respectively. Then, the balanced dataset $\mathcal{D}_a^{\text{bal}}$ for user $a$ is $\mathcal{D}_a^{\text{bal}} = \mathcal{D}_a^g \cup \mathcal{O}_a^g \cup \mathcal{D}_a^f \cup \mathcal{O}_a^f$, The complete balanced dataset $\mathcal{D}^{\text{bal}}$ of the client is the union of the balanced datasets of all users $\mathcal{D}^{\text{bal}} = \bigcup_{a \in \mathcal{A}} \mathcal{D}_a^{\text{bal}}$, This process ensures that the number of genuine signatures and forged signatures for each author is equal, thereby mitigating the impact of class imbalance during model training.

To enhance the robustness of the balanced dataset $\mathcal{D}^{\text{bal}}$, all samples $x$ in $\mathcal{D}^{\text{bal}}$ undergo a hybrid augmentation strategy $\mathcal{T}$. This transformation consists of a random sequence of geometric and non-geometric sample operations. For an input sample $x$, the augmented sample $x'$ is expressed as,

$$x' = \mathcal{T}_j(x), \tag{3}$$

where $\mathcal{T}_j$ denotes the $j$-th sub-transformation function. The detailed description and selection method of the sub-transformation functions are specified in **Appendix A**. The final augmented dataset $\mathcal{D}^{\text{aug}}$ is $\mathcal{D}^{\text{aug}} = \{(\mathcal{T}(x), y^c, y^v) \mid (x, y^c, y^v) \in \mathcal{D}^{\text{bal}}\}$.

### 3.2 DUAL-TASK HEAD MECHANISM

In the scenario of federated signature detection, the local client model needs to handle two tasks simultaneously: user classification and authenticity detection. We define their loss functions as follows,

$$\mathcal{L}_c(\theta) = \frac{1}{N} \sum_{i \in \mathcal{D}^{\text{aug}}}^{N} \ell_c(f_c(\theta; x_i), y_i^c), \quad \mathcal{L}_v(\theta) = \frac{1}{N} \sum_{i \in \mathcal{D}^{\text{aug}}}^{N} \ell_v(f_v(\theta; x_i), y_i^v), \tag{4}$$

where $\theta$ denotes the parameters of the feature extractor, and $N$ represents the total number of samples in the augmented dataset $\mathcal{D}^{\text{aug}}$. The classification head $f_c(\theta; x_i)$ outputs a prediction, and the cross-entropy loss $\ell_c$ calculates the discrepancy between this prediction and the genuine user category label $y_i^c$. Simultaneously, the verification head $f_v(\theta; x_i)$ outputs predictions, and the binary cross-entropy loss $\ell_v$ calculates the discrepancy between these predictions and the authenticity label $y_i^v$. Finally, the user classification loss $\mathcal{L}_c(\theta)$ and the authenticity verification loss $\mathcal{L}_v(\theta)$ are obtained by accumulating and averaging the sample losses, respectively.

The optimization goal of the module is to coordinate the two task heads to update parameters until convergence under the guidance of Pareto optimization. To achieve this goal, a weighted sum approach is used to construct the overall optimization objective, specifically as follows,

$$\mathcal{L}(\theta; \lambda) = (1 - \lambda)\mathcal{L}_c(\theta) + \lambda\mathcal{L}_v(\theta), \tag{5}$$

where $\mathcal{L}(\theta; \lambda)$ denotes the weighted total loss function. By adjusting the hyperparameter $\lambda$ to balance the losses of the two tasks, the model is guided toward optimizing toward a Pareto optimal solution.

The core challenge of the optimization objective lies in dynamically determining the optimal weight $\lambda^*$ during each parameter update iteration, rather than relying on a fixed value, which is essential for task balance. According to the Pareto stationary condition, at the optimal solution $\theta^*$, there exists $\lambda^* \in [0, 1]$ such that the combined gradient $(1 - \lambda^*)\nabla\mathcal{L}_c(\theta^*) + \lambda^*\nabla\mathcal{L}_v(\theta^*) = 0$.

The weight that minimizes the norm of the combined gradient can be found by solving the convex optimization problem,

$$\min_{\lambda \in [0,1]} \|(1 - \lambda)\nabla_c + \lambda\nabla_v\|_2^2, \tag{6}$$

where $\nabla_c = \nabla L_c(\theta)$ and $\nabla_v = \nabla L_v(\theta)$. Finally, the solution obtained is the following,

$$\lambda^* = \frac{\langle \nabla_c, \nabla_c - \nabla_v \rangle}{\|\nabla_c - \nabla_v\|_2^2}, \tag{7}$$

and the detailed solution process is shown in **Appendix B**. It is worth noting that to ensure the effectiveness of $\lambda^*$, the calculated $\lambda^*$ is subsequently truncated to $[0, 1]$. Based on the calculated optimal weights, the gradients are combined as $\nabla_t = (1 - \lambda_t^*)\nabla_c + \lambda_t^*\nabla_v$, and the parameter update $\theta_{t+1} = \theta_t - \eta\nabla_t$ is performed. This update rule ensures that the optimization direction moves along the tangential direction of the Pareto frontier.

### 3.3 EFFICIENT PARETO GRADIENT OPTIMIZATION

To address the challenge of multiple task heads within clients in a heterogeneous model environment, we propose an efficient Pareto gradient optimization protocol. By efficiently evaluating a carefully selected set of high-potential gradient update combinations and stopping the search upon meeting the requirements, we avoid computationally expensive exhaustive searches.

Specifically, after the server collects the gradient updates uploaded by the clients, it groups them by task type, the gradient set for classification tasks $\{g_c^{(i)}\}_{i=0}^{n-1}$ and the gradient set for detection tasks $\{g_v^{(j)}\}_{j=0}^{n-1}$, where $n$ represents the number of clients participating in the current training. Then, the collected gradients are accumulated to obtain the linear gradient combination used in optimization, as shown in the following equation.

$$G_c^{(m)} = \sum_{k=0}^{m} g_c^{(k)} \quad m \in \{0, 1, \dots, n-1\}, \quad G_v^{(m)} = \sum_{k=0}^{m} g_v^{(k)} \quad m \in \{0, 1, \dots, n-1\}, \tag{8}$$

where $n$ is the length of the gradient list and the number of clients participating in the training. $g_c^{(k)}$ and $g_v^{(k)}$ represent the $k$-th gradient in the classification gradient set and the detection gradient set, respectively. $G_c^{(m)}$ and $G_v^{(m)}$ represent the cumulative result of classification gradients and the cumulative result of detection gradients up to the $m$-th gradient, respectively.

The ideal goal of the module is to find the optimal combination of update steps $(\alpha^*, \beta^*)$, where $\alpha, \beta \in [0, n)$, such that after applying the corresponding accumulated gradients, the new model minimizes the overall loss while satisfying the Pareto constraints, specifically as follows.

$$(\alpha^*, \beta^*) = \arg\min_{\alpha, \beta} \left( \mathcal{L}_c(\theta_c - G_c^{(\alpha)}) + \mathcal{L}_v(\theta_v - G_v^{(\beta)}) \right). \tag{9}$$

where $\theta_c$ is the parameter of the classification head, and $\theta_v$ is the parameter of the detection task head. To obtain $(\alpha^*, \beta^*)$, we search for the most likely combinations from all possible binary pairs. The specific method is as follows.

$$S_{\text{opt}} = S_{\text{sync}} \cup S_{\text{boundary}} \cup S_{\text{bias}} \tag{10}$$

where

$$\begin{aligned}
S_{\text{sync}} &= \{(i, i) \mid i = 1, \dots, n-1\}, \\
S_{\text{boundary}} &= \{(n-1, 0), (0, n-1)\}, \\
S_{\text{bias}} &= \begin{cases} \emptyset & \text{if } n < 3, \\ \{(n-1, \lfloor n/2 \rfloor), (\lfloor n/2 \rfloor, n-1)\} & \text{if } n \geq 3. \end{cases}
\end{aligned} \tag{11}$$

where $S_{\text{sync}}$ is the synchronous optimization set, representing combinations where the step counts for both tasks are equal. $S_{\text{boundary}}$ is the boundary extremum set, representing the boundary combinations where a single task reaches the maximum number of steps. $S_{\text{bias}}$ is the bias exploration set, representing the biased combinations with asymmetric numbers of steps between tasks.

These three sets represent the most promising gradient combinations for efficiency and effectiveness. The $S_{\text{sync}}$ set uses a synchronous update strategy, akin to standard single-task federated learning, treating both task heads equally for computational savings. The $S_{\text{boundary}}$ set leverages larger step sizes for greater client gradient aggregation, which counters gradient vanishing near convergence by incorporating more contributions. Finally, given the task difficulty gap in offline signature verification (binary forgery detection vs. user classification), the $S_{\text{bias}}$ set's asymmetric exploration adapts to these imbalances, enhancing overall performance. Experiments in Appendix E validate this selection.

To satisfy the Pareto constraint, we need to perform a Pareto improvement check when updating the loss. Specifically, it is the Pareto dominance test,

$$\mathcal{L}_c(\theta_c - G_c^{(\alpha^*)}) \leq \mathcal{L}_c(\theta_c) + \varepsilon \quad \text{and} \quad \mathcal{L}_v(\theta_v - G_v^{(\beta^*)}) \leq \mathcal{L}_v(\theta_v) + \varepsilon, \tag{12}$$

where $\varepsilon > 0$ is a numerical tolerance parameter (typically set to $10^{-6}$) used to compensate for floating-point calculation errors. This condition ensures that the new solution is not inferior to the benchmark solution, satisfying the basic requirements of Pareto improvement. Next, it must undergo a significant improvement test,

$$\frac{\mathcal{L}_c(\theta_c) - \mathcal{L}_c(\theta_c - G_c^{(\alpha^*)})}{|\mathcal{L}_c(\theta_c)| + \delta} > \vartheta \quad \text{or} \quad \frac{\mathcal{L}_v(\theta_v) - \mathcal{L}_v(\theta_v - G_v^{(\beta^*)})}{|\mathcal{L}_v(\theta_v)| + \delta} > \vartheta, \tag{13}$$

where $\vartheta > 0$ is a predefined significance threshold. This condition ensures that at least one objective function achieves a relative performance improvement exceeding the threshold $\vartheta$. Finally, an overall performance non-degeneration test must be conducted,

$$\mathcal{L}_c(\theta_c - G_c^{(\alpha^*)}) + \mathcal{L}_v(\theta_v - G_v^{(\beta^*)}) < \mathcal{L}_c(\theta_c) + \mathcal{L}_v(\theta_v) \tag{14}$$

To save computational resources and terminate ineffective aggregation operations that may lead to performance degradation. After meeting the above three inspection requirements, the search will not continue. The overall method flow is shown in Algorithm 1.

### 3.4 CONVERGENCE ANALYSIS

In this subsection, this paper provides a theoretical analysis of the convergence of FedMH. The focus of the convergence analysis of FedMH is to prove that the proposed framework converges to a Pareto stationary point under standard assumptions in the scenario of federated handwritten signature verification. We mainly consider the non-convex setting, which aligns with the requirements of conventional signature verification scenarios. Detailed proofs are provided in **Appendix C**. The specific assumptions are as follows (Tan et al., 2022; Li et al., 2020; Karimireddy et al., 2020; Chan et al., 2024).

**Assumption 1.** ($\beta$-smoothness) *The task loss functions $L_c(\theta)$ and $L_v(\theta)$ in Equation 4 are $\beta$-smooth. That is, for any $\theta_1, \theta_2$ in the domain, the following condition holds $\|\nabla\mathcal{L}(\theta_1) - \nabla\mathcal{L}(\theta_2)\| \leq \beta\|\theta_1 - \theta_2\|$.*

**Assumption 2.** (Bounded Gradient) *The gradient of the loss function is bounded. There exists a constant $H > 0$ such that for any $\theta$, it holds that $\|\nabla\mathcal{L}(\theta)\| \leq H$.*

**Assumption 3.** (Unbiased stochastic gradients with bounded variance) *The local stochastic gradients computed from mini-batches of the augmented dataset $D^{aug}$ are unbiased estimates of the true gradients. Moreover, their variances are bounded by $\sigma^2$. Formally, for any $\theta$, $\mathbb{E}[g(\theta)] = \nabla\mathcal{L}(\theta), \mathbb{E}[\|g(\theta) - \nabla\mathcal{L}(\theta)\|^2] \leq \sigma^2$.*

Meanwhile, according to the Pareto test in this paper, Lemma 1 can also be obtained.

**Lemma 1** (Property of Pareto Improvement Update). *The gradient optimization direction $\mathbf{G}_t$ selected by the FedMH method satisfies the following condition, where there exists a constant $\gamma \geq \frac{\beta}{2}$ such that $\langle\nabla\mathcal{L}_{t,\Upsilon}, \mathbf{G}_t\rangle \geq \gamma\|\mathbf{G}_t\|^2$.*

Under these assumptions and based on Lemma 1, we establish the following convergence results for the non-convex case.

**Theorem 1** (Main Convergence Result). *Based on Assumptions 1-3 and Lemma 1, the model parameter $\theta_{t,e}$ trained by the proposed algorithm satisfies the following inequality after $T$ rounds of*

---

**Algorithm 1** Overall Process of the FedMH Method

---

**Require:** Rounds $T$, clients $\mathcal{C}$, local epochs $\Upsilon$, learning rate $\eta$
1: Initialize $\mathcal{M}_0 = \{f_\phi, h_c, h_v\}$
2: **for** client $k \in \mathcal{C}$ **in parallel do**
3:    $\mathcal{M}_k \leftarrow \mathcal{M}_0$
4: **end for**
5: **for** $t = 1$ to $T$ **do**
6:    **for** client $k \in \mathcal{C}$ **in parallel do**
7:       $\mathcal{D}'_k \leftarrow \textsc{Preprocess}(\mathcal{D}_k)$ {Balance & augment}
8:       **for** $e = 1$ to $\Upsilon$ **do**
9:          **for** batch $(X, y_c, y_v) \in \mathcal{D}'_k$ **do**
10:            $z \leftarrow f_\phi(X)$
11:            $\mathcal{L}_c \leftarrow \ell_c(h_c(z), y_c, y_v = 1), \quad \mathcal{L}_v \leftarrow \ell_v(h_v(z), y_v)$
12:            $g_c \leftarrow \nabla_\theta \mathcal{L}_c, \quad g_v \leftarrow \nabla_\theta \mathcal{L}_v, \quad \lambda \leftarrow \min\left(\max\left(\frac{\langle g_c, g_c - g_v \rangle}{\|g_c - g_v\|^2}, 0\right), 1\right)$
13:            $\theta \leftarrow \theta - \eta[(1 - \lambda)g_c + \lambda g_v]$
14:          **end for**
15:       **end for**
16:       Send $\Delta_k = \{\Delta_c^k, \Delta_v^k\}$ to Server
17:    **end for**
18:    **Server aggregation:**
19:    $G_c \leftarrow \textsc{CumulativeSum}(\{\Delta_c^k\}_{k \in \mathcal{S}_t}), G_v \leftarrow \textsc{CumulativeSum}(\{\Delta_v^k\}_{k \in \mathcal{S}_t})$
20:    Send $\{G_c, G_v\}$ to clients
21:    **for** client $k \in \mathcal{C}$ **in parallel do**
22:       $(\alpha^*, \beta^*) \leftarrow \textsc{ParetoSearch}(G_c, G_v, \mathcal{M}_t)$
23:       $\mathcal{M}'_k = \mathcal{M}_k - (G_c^{\alpha^*}, G_v^{\beta^*})$
24:    **end for**
25: **end for**
26: **return** Client Feature Extractor Model $\{\mathcal{M}_1^f, \ldots, \mathcal{M}_n^f\}$

---

*communication:*

$$\min_{t \in \{0, \ldots, T\}, e \in \{0, \ldots, \Upsilon\}} \mathbb{E}[\|\nabla \mathcal{L}(\theta_{t,e}; \lambda^*)\|^2] \leq \frac{\mathcal{L}_0 - \mathcal{L}^* + \frac{\beta \eta^2 \Upsilon \sigma^2}{2} T - \tilde{\kappa} n^2 H^2 T}{\Upsilon T \left(\eta - \frac{\beta \eta^2}{2}\right)},$$

where $\mathcal{L}_0 = \mathcal{L}(\theta_{0,0}; \lambda^*)$, $\mathcal{L}^*$ is the lower bound of the loss function, and $\tilde{\kappa} = \gamma - \frac{\beta}{2} \geq 0$. Furthermore, if the choice of the learning rate $\eta$ satisfies $\eta < \frac{2}{\beta}$ and $\tilde{\kappa} > 0$, the algorithm converges to a stationary point at a rate of $\mathcal{O}(1/T)$.

## 4 EXPERIMENT

### 4.1 EXPERIMENTAL CONFIGURATION

**Dataset:** Experiments are conducted on the **GPDS-1000 (Ferrer et al., 2016), Bengali, Hindi (Pal et al., 2016) and CEDAR (Kalera et al., 2004)** datasets to evaluate the performance of the proposed FedMH. Both the Bengali and Hindi subsets are derived from the BHSig260 corpus.

**Performance Metrics:** To evaluate the verification performance of the proposed method, the following metrics are taken into account. **AUC** is the Area under the ROC curve. **FRR** is the rate at which genuine signatures are incorrectly classified as forgeries. **FAR** is the ratio of skilled forgeries that are incorrectly validated as genuine signatures. **Equal Error Rate (EER)** is the error rate when FAR = FRR.

**Training settings:** Batch size is 32, local learning rate is 0.01, with 100 global training rounds and 2 local epochs per round. Feature dimension is 2048. We use 700 out of 1000 GPDS users, with signatures distributed via the Dirichlet distribution ($\alpha = 0.1$) to simulate heterogeneity. The experiment involves the SigNet (Hafemann et al., 2017), ViTs (Dosovitskiy et al., 2021), HTCSigNet (Zheng

Table 1: Performance comparison of different federated learning methods under the setup with 5 homogeneous client models and the GPDS-700 dataset. **Bold** indicates the best performance for each metric.

| Method | OctConvNet | | | | SigNet | | | | ViTs | | | | HTCSignet | | | |
|---|---|---|---|---|---|---|---|---|---|---|---|---|---|---|---|---|
| | EER | AUC | FAR | FRR | EER | AUC | FAR | FRR | EER | AUC | FAR | FRR | EER | AUC | FAR | FRR |
| FedProto | 22.7 | 82.6 | 23.2 | 22.2 | 13.6 | 90.8 | 13.9 | 13.3 | 22.7 | 82.5 | 23.2 | 22.1 | 25.78 | 78.28 | 26.36 | 25.20 |
| LG-FedAvg | 15.0 | 90.0 | 15.2 | 14.7 | 9.2 | 94.2 | 9.4 | 8.9 | 23.5 | 81.4 | 24.2 | 22.7 | 21.66 | 83.42 | 22.20 | 21.12 |
| FML | 14.6 | 90.1 | 15.0 | 14.1 | 8.2 | 94.6 | 8.5 | 8.0 | 23.3 | 81.8 | 23.9 | 22.7 | 21.76 | 83.31 | 22.24 | 21.27 |
| FedKD | 18.8 | 85.8 | 19.6 | 18.0 | 18.4 | 85.8 | 19.2 | 17.6 | 21.4 | 83.5 | 21.8 | 20.9 | 11.83 | 92.03 | 12.36 | 11.30 |
| FedGH | 14.4 | 90.2 | 14.9 | 14.0 | 9.2 | 94.0 | 9.5 | 8.9 | 22.7 | 82.1 | 23.4 | 22.1 | 21.80 | 83.35 | 22.35 | 21.25 |
| FedGen | 15.7 | 89.1 | 16.1 | 15.3 | 8.0 | 94.6 | 8.4 | 7.7 | 23.9 | 81.2 | 24.6 | 23.1 | 16.97 | 87.75 | 17.36 | 16.58 |
| FedTGP | 21.6 | 83.6 | 22.0 | 21.1 | 16.1 | 88.3 | 16.4 | 15.7 | 28.2 | 77.2 | 28.9 | 27.6 | 18.09 | 87.30 | 18.43 | 17.75 |
| FedMH | **9.6** | **93.6** | **9.8** | **9.3** | **5.3** | **96.9** | **5.5** | **5.1** | **17.0** | **87.3** | **17.3** | **16.7** | **7.99** | **94.83** | **8.34** | **7.65** |

Table 2: Performance comparison of different federated learning methods under the setup with 5 client **heterogeneous** client models and the GPDS-700 dataset. **Bold** indicates the best performance for each metric.

| Method | HOctConvNet | | | | HSigNet | | | | HViTs | | | |
|---|---|---|---|---|---|---|---|---|---|---|---|---|
| | EER | AUC | FAR | FRR | EER | AUC | FAR | FRR | EER | AUC | FAR | FRR |
| FedProto | 16.8 | 87.7 | 17.1 | 16.6 | 14.2 | 90.3 | 14.6 | 13.8 | 25.1 | 80.1 | 25.8 | 24.4 |
| LG-FedAvg | 10.2 | 93.4 | 10.3 | 10.0 | 8.2 | 94.5 | 8.4 | 8.0 | 22.5 | 82.8 | 23.2 | 21.9 |
| FML | 10.8 | 93.0 | 11.3 | 10.3 | 8.3 | 94.6 | 8.6 | 8.0 | 23.3 | 82.3 | 23.9 | 22.6 |
| FedKD | 17.9 | 86.3 | 18.5 | 17.3 | 17.4 | 87.1 | 18.0 | 16.7 | 22.4 | 82.8 | 22.9 | 21.9 |
| FedGH | 10.7 | 93.0 | 10.9 | 10.4 | 8.9 | 94.1 | 9.1 | 8.7 | 23.4 | 82.0 | 24.1 | 22.7 |
| FedGen | 10.5 | 92.9 | 10.8 | 10.2 | 7.7 | 94.7 | 7.9 | 7.5 | 22.8 | 82.5 | 23.4 | 22.3 |
| FedTGP | 15.9 | 88.4 | 16.2 | 15.7 | 16.1 | 88.6 | 16.5 | 15.8 | 26.6 | 78.1 | 27.1 | 26.0 |
| FedMH | **7.4** | **95.1** | **7.5** | **7.2** | **5.1** | **96.9** | **5.3** | **4.9** | **17.7** | **86.8** | **18.1** | **17.4** |

et al., 2025) and OctConvNet (Gutub et al., 2025) models. The heterogeneous model is set as follows. HSigNet uses SigNet and SigNet_S; HOctConvNet uses OctConvNet and SigNet; HViTs uses ViTSigNet and ViTSigNet-32.

**Test settings:** Writer-Dependent SVM evaluates feature quality. Generalization uses SVM with RBF kernel for five-fold cross-validation, $224 \times 224$ pixel images, 12 genuine signatures for training, 10 for testing, 12 forged as negative samples, batch size 32. Personalization uses 700 GPDS users, 30% as test set, same users for training or testing. The generalization ability evaluation uses the client model after 100 rounds; other evaluations use the client model from the last 10 rounds of global training. For details on the settings, please refer to **Appendix D**.

## 4.2 BASELINE PERFORMANCE COMPARISON EXPERIMENT

Tables 1 and 2 show the personalized performance of the FedMH method. The test scenario simulates the need for identifying skilled forged signatures for their own user groups in different institutions. Specifically, the experiment uses 700 user signature samples from GPDS as the dataset, with 30% selected as the test set, and the results are the average performance of the model after the last 10 rounds of communication. The results show that FedMH effectively handles both homogeneous and heterogeneous client model settings, with a more pronounced advantage in the heterogeneous case, as it leverages more effective gradient updates from diverse model combinations. This is because the FedMH method can obtain more effective update gradients from different model combinations.

In the generalization experiment results, the experiment mainly demonstrates the performance of the method when dealing with unfamiliar users. Therefore, the model trained by 5 clients on the dataset of 700 users from GPDS is used for testing on unfamiliar users and datasets with signatures in different languages. The results are shown in Figure 2. It can be seen that FedMH has the greatest advantage in the test on unfamiliar users with data in the same language. In the dataset with signatures in different languages, although the performance advantage of FedMH is reduced, it is still better than other methods.

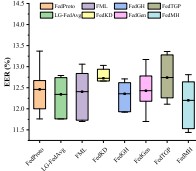 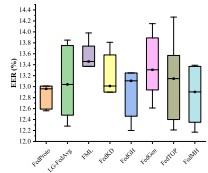 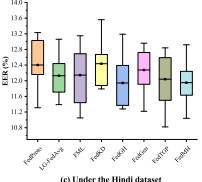 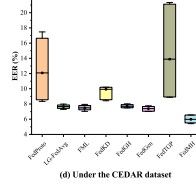

(a) Under the GPDS-300 user dataset  (b) Under the Bengali dataset  (c) Under the Hindi dataset  (d) Under the CEDAR dataset

Figure 2: Experimental results on generalization performance under three datasets and the heterogeneous model setting **HSigNet**, where the **average EER (%)** of each federated learning method is highlighted.

Table 3: Results of the ablation experiments using the SigNet model and the GPDS-700 dataset. Where **DA** refers to the data augmentation module, **DH** is the dual-task head mechanism, and **GO** refers to the gradient optimization module.

| Module Configuration | | | Performance indicators (%) | | | |
|:---:|:---:|:---:|:---:|:---:|:---:|:---:|
| **DA** | **DH** | **GO** | **EER** | **AUC** | **FAR** | **FRR** |
| ✗ | ✗ | ✗ | $8.48 \pm 0.01$ | $94.41 \pm 0.00$ | $8.69 \pm 0.01$ | $8.28 \pm 0.00$ |
| ✗ | ✓ | ✗ | $7.75 \pm 0.21$ | $95.13 \pm 0.21$ | $8.00 \pm 0.22$ | $7.49 \pm 0.21$ |
| ✓ | ✗ | ✗ | $5.58 \pm 0.16$ | $96.74 \pm 0.13$ | $5.82 \pm 0.17$ | $5.34 \pm 0.16$ |
| ✓ | ✓ | ✗ | $5.54 \pm 0.25$ | $96.70 \pm 0.16$ | $5.74 \pm 0.25$ | $5.33 \pm 0.24$ |
| ✓ | ✓ | ✓ | $\mathbf{5.28 \pm 0.17}$ | $\mathbf{96.91 \pm 0.13}$ | $\mathbf{5.47 \pm 0.18}$ | $\mathbf{5.09 \pm 0.17}$ |

## 4.3 ABLATION STUDY AND COMPONENT ANALYSIS

To verify the effectiveness of each component in the proposed DA, DH, and GO modules, we conducted comprehensive ablation studies. First, as shown in Table 3, under the homogeneous setting, the DA module achieved a significant improvement in baseline performance, reducing the EER from $8.48 \pm 0.01$ to $5.58 \pm 0.16$. This indicates that the DA module plays a crucial role in addressing the imbalance of local client data and enhancing samples. On this basis, the DH and GO modules further reduced the EER to $5.28 \pm 0.17$, providing valuable incremental improvements. However, the true value of the DH and GO modules is fully demonstrated when dealing with model heterogeneity. We present supplementary ablation experiments under the heterogeneous setting in Table 11. In the heterogeneous scenario, the marginal contribution of the DH and GO modules increases significantly. The DA module alone reduces the EER from $9.96 \pm 0.46$ to $8.05 \pm 0.35$. After adding the DH and GO modules, the EER is further reduced to $7.43 \pm 0.62$, resulting in an additional EER reduction of 0.62%. Compared with the 0.3% marginal gain in the homogeneous setting ($5.58 \rightarrow 5.28$), the gain in the heterogeneous setting (0.62%) is more than twice as large. This comparison strongly demonstrates that while DA is the foundation for addressing data imbalance, DH and GO are the core mechanisms of the FedMH framework that effectively coordinate multi-tasks and aggregate knowledge in a heterogeneous model environment.

Figure 3 visually presents, through a heatmap, the frequency with which different gradient combinations $(i, j)$ in the efficient Pareto gradient optimization protocol are selected under different improvement thresholds $\vartheta$ in a setup with 5 clients. The heatmap clearly shows that regardless of how $\vartheta$ is set, the combination $(i = 4, j = 0)$ is selected with an absolute advantage (for example, 86 times when $\vartheta = 0.001$). This combination $(n - 1, 0)$ corresponds to the boundary extreme set $S_{\text{boundary}}$. This figure strongly validates the effectiveness of the $S_{\text{opt}}$ search space strategy, as the most frequently selected combination in the experiment is successfully captured by $S_{\text{opt}}$, thereby supporting that this strategy effectively avoids the $O(n^2)$ exhaustive search while ensuring performance. It is worth noting that the gradient combinations selected in the early stage of training are often the $S_{\text{sync}}$ synchronous optimization set, and the gradient combinations selected later are the $S_{\text{boundary}}$ boundary extreme set. This indicates that in the early stage of training, the user classification task and the authenticity identification task can collaborate with each other, and later on, the gradients generated by the authenticity identification task often meet the Pareto verification requirements. This is consistent with the situation where the authenticity signature identification task is dominant in offline signature identification tasks. The results in Table 10 show that the time complexity of the $S_{opt}$

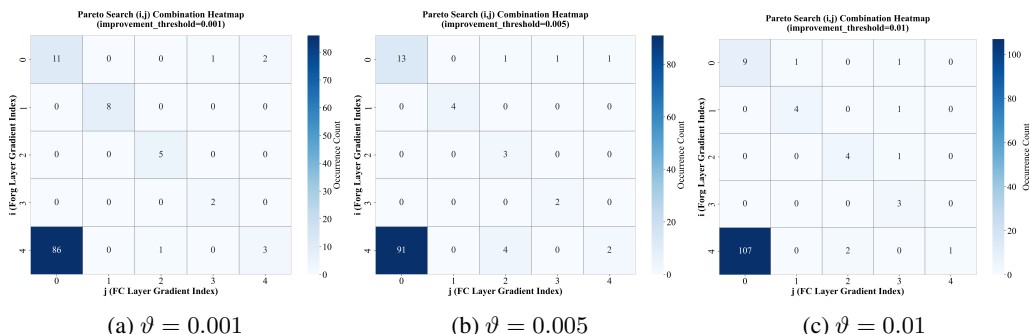

(a) $\vartheta = 0.001$  (b) $\vartheta = 0.005$  (c) $\vartheta = 0.01$

Figure 3: Pareto gradient selection frequency analysis ($n = 5$). The heatmap shows the selection frequency of gradient combinations (i, j) under different thresholds $\vartheta$.

strategy is only $O(n)$, which is much lower than the $O(n^2)$ complexity of the $S_{all}$ exhaustive search. However, in terms of performance, $S_{opt}$ achieved an EER of $5.28 \pm 0.17$, which is not only better than the baseline without using the GO module (EER $5.54 \pm 0.25$) but even significantly better than the $S_{all}$ strategy (EER $5.78 \pm 0.12$). This confirms that the gradient set selected by the GO module is not only efficient but also has better performance. This is because $S_{opt}$ prioritizes testing the gradient combinations that are most likely to achieve the best optimization, thereby searching for a path more conducive to stable convergence. On the contrary, $S_{all}$ selects the instantaneous optimal, that is, the gradient combination that meets the $\vartheta$ threshold but has worse long-term convergence.

To demonstrate the role of the DA module, we analyze its key strategies in Figure 4. The experimental results clearly show that the **ALL** configuration (combining oversampling and transformation) achieves the lowest and most stable EER. In contrast, using the **Oversampling** strategy alone results in the highest and most dispersed EER, indicating that oversampling alone can lead to overfitting. Although the **Transformation** strategy outperforms the **Non** baseline, its performance is still far inferior to the **ALL** configuration. This confirms that our combined strategy is necessary.

The effectiveness of the DH model is verified in Figure 5. Figure 5(a) shows that **ALL** (the complete DH mechanism) outperforms both the **Non** (single-task head) setup and the **Sequential** (sequential training) setup throughout the entire training process. Notably, the **Sequential** mode begins to overfit after the 70th round, exposing the issue of gradient conflicts in later stages of model training caused by sequential execution. In addition, Figure 5(b) and (c) present a comparison between our dynamic $\lambda$ weight strategy and multiple static $\lambda$ weights. All static weight setups exhibit an overfitting trend in the later stages of training (approximately 90-100 rounds), while the dynamic $\lambda$ mechanism can flexibly adjust the focus of tasks, further optimize the model, and successfully avoid performance degradation. This demonstrates that dynamic weight adjustment is crucial for balancing dual tasks and achieving robust convergence. For more experiments, please refer to Appendix E.

## 5 CONCLUSION

This paper proposes a novel method called FedMH for the application of heterogeneous federated learning in the scenario of offline handwritten signature verification. This method effectively addresses the relevant challenges in the scenario of offline handwritten signature verification through three modules: an adaptive enhancement strategy, a dual-task head mechanism, and a gradient optimization protocol, guided by the Pareto improvement theory, and provides a corresponding theoretical basis. Experiments show that FedMH outperforms some current heterogeneous federated learning baselines under various settings. However, it is worth noting that the FedMH method still needs improvement in terms of convergence speed, computational overhead, and communication overhead. This will be the focus of our future work.

## 6 REPRODUCIBILITY STATEMENT

The code for the method implementation can be found in the anonymous repository at the link `https://anonymous.4open.science/r/FedMH-13BF`. Additionally, we have included the final model parameters in the supplementary materials.

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

# A  DATA TRANSFORMATION FUNCTION

This section elaborates on the data transformation functions used by FedMH, which are divided into two categories: geometric transformations and non-geometric transformations. In addition, this section also provides a method for extracting transformation functions, which uses a polynomial probability distribution to control the use of transformation functions.

## A.1  TRANSFORMATION FUNCTIONS

Data transformation functions are divided into two categories: one is geometric transformation, which changes the spatial structure of pixels in image samples; the other is non-geometric transformation, which modifies the intensity of pixels in image samples or introduces noise and blur effects. These functions will be described in detail in Table 4 below.

## A.2  TRANSFORMATION SAMPLING STRATEGY

The use of the transformation function follows a probability sampling strategy based on the polynomial probability distribution.

First, let the list of $M$ predefined data transformation function schemes be $\mathcal{F} = \{F_1, F_2, \ldots, F_M\}$, where each scheme $F_i = \{(\mathcal{T}_i^{(1)}, p_i^{(1)}), (\mathcal{T}_i^{(2)}, p_i^{(2)}), \ldots, (\mathcal{T}_i^{(K)}, p_i^{(K)})\}$ is a list containing $K$ tuples. Each tuple $(\mathcal{T}_i^{(j)}, p_i^{(j)})$ specifies a transformation $\mathcal{T}_i^{(j)}$ and its application probability $p_i^{(j)} \in [0, 1]$. Then, a solution $F_i$ is selected uniformly at random from $\mathcal{F}$, with its probability as shown below

$$p(F_i) = \frac{1}{M}, \quad \forall i = 1, 2, \ldots, M.$$

For each tuple $(\mathcal{T}_i^{(j)}, p_i^{(j)})$ in the selected strategy $F_i$, the transformation function $\mathcal{T}_i^{(j)}$ will be used with a probability of $p_i^{(j)}$, which follows a Bernoulli distribution, specifically

$$p(\text{use } \mathcal{T}_i^{(j)}) = p_i^{(j)}, \quad p(\text{Not use } \mathcal{T}_i^{(j)}) = 1 - p_i^{(j)}.$$

The use of each transformation function is determined by a random variable $X_i^{(j)} \sim \text{Bernoulli}(p_i^{(j)})$, where $X_i^{(j)} = 1$ indicates the use of the transformation function $\mathcal{T}_i^{(j)}$, and $X_i^{(j)} = 0$ indicates that the transformation function is not used.

Table 4: Overview of Image Transformation Methods

| Category | Transformation | Description |
|---|---|---|
| **Geometric Transformations** | **Shear** | Apply a shear transformation, which shears along the horizontal or vertical axis according to the set shear strength. |
| | **Translation** | Shift the image according to the specified displacement. |
| | **Rotation** | Rotate the image at an angle within the range. |
| | **Elastic** | Introduce elastic deformation through a displacement field generated by Gaussian smoothing to simulate local distortion. |
| **Non-Geometric Transformations** | **Adjust Contrast** | Modify the image contrast according to the specified range of contrast adjustment. |
| | **Adjust Brightness** | Change the image brightness by defining the intensity of brightness change. |
| | **Adjust Gamma** | Set the range of gamma values and apply gamma correction. |
| | **Gaussian Noise** | Introduce Gaussian noise to simulate sensor noise. |
| | **Gaussian Blur** | Apply Gaussian blur for smoothing effects. |
| | **Motion Blur** | Simulate directional blur to mimic motion effects. |
| | **Cropping** | Randomly remove rectangular blocks from the image. |
| | **Mixed Noise** | Blend the image with noise patterns or another image. |

The use or non-use of various transformation functions in a scheme forms a polynomial probability distribution. For a scheme $F_i$ containing $K$ types of transformations, the probability of applying a specific subset of transformations $\mathcal{S} \subseteq \{\mathcal{T}_i^{(1)}, \mathcal{T}_i^{(2)}, \ldots, \mathcal{T}_i^{(K)}\}$ is:

$$p(\mathcal{S}) = \prod_{\mathcal{T}_i^{(j)} \in \mathcal{S}} p_i^{(j)} \cdot \prod_{\mathcal{T}_i^{(j)} \notin \mathcal{S}} (1 - p_i^{(j)}).$$

This probabilistic framework enables a flexible and controlled augmentation pipeline, enhancing model generalization while maintaining computational tractability.

## B  DERIVATION OF OPTIMAL WEIGHT $\lambda^*$

The derivation process for the optimal weight $\lambda^*$ is detailed as follows. First, the corresponding optimization problem is formulated as

$$\min_{\lambda \in [0,1]} \|(1 - \lambda)\nabla_c + \lambda \nabla_v\|_2^2,$$

where $\nabla_c = \nabla_\theta L_c(\theta)$ and $\nabla_v = \nabla_\theta L_v(\theta)$.

Let $\mathbf{g}_c = \nabla_c$ and $\mathbf{g}_v = \nabla_v$. The objective function is

$$f(\lambda) = \|(1 - \lambda)\mathbf{g}_c + \lambda \mathbf{g}_v\|_2^2 = \|\mathbf{g}_c + \lambda(\mathbf{g}_v - \mathbf{g}_c)\|_2^2.$$

Define $\mathbf{g}_d = \mathbf{g}_v - \mathbf{g}_c$. Then,

$$f(\lambda) = \|\mathbf{g}_c + \lambda \mathbf{g}_d\|_2^2 = \langle \mathbf{g}_c + \lambda \mathbf{g}_d, \mathbf{g}_c + \lambda \mathbf{g}_d \rangle = \|\mathbf{g}_c\|_2^2 + 2\lambda \langle \mathbf{g}_c, \mathbf{g}_d \rangle + \lambda^2 \|\mathbf{g}_d\|_2^2.$$

This is a quadratic function in $\lambda$:

$$f(\lambda) = \alpha \lambda^2 + \beta \lambda + \gamma,$$

where $\alpha = \|\mathbf{g}_d\|_2^2$, $\beta = 2\langle \mathbf{g}_c, \mathbf{g}_d \rangle$, and $\gamma = \|\mathbf{g}_c\|_2^2$. Assuming $\mathbf{g}_d \neq \mathbf{0}$, we have $\alpha > 0$, so $f(\lambda)$ is strictly convex.

To find the unconstrained minimizer, compute the derivative:

$$f'(\lambda) = 2\alpha\lambda + \beta = 2\|\mathbf{g}_d\|_2^2 \cdot \lambda + 2\langle \mathbf{g}_c, \mathbf{g}_d \rangle.$$

Set $f'(\lambda) = 0$:

$$2\|\mathbf{g}_d\|_2^2 \cdot \lambda + 2\langle \mathbf{g}_c, \mathbf{g}_d \rangle = 0 \implies \lambda = -\frac{\langle \mathbf{g}_c, \mathbf{g}_d \rangle}{\|\mathbf{g}_d\|_2^2}.$$

Substitute $\mathbf{g}_d = \mathbf{g}_v - \mathbf{g}_c$:

$$\langle \mathbf{g}_c, \mathbf{g}_d \rangle = \langle \mathbf{g}_c, \mathbf{g}_v - \mathbf{g}_c \rangle = \langle \mathbf{g}_c, \mathbf{g}_v \rangle - \|\mathbf{g}_c\|_2^2,$$

so

$$\lambda = -\frac{\langle \mathbf{g}_c, \mathbf{g}_v \rangle - \|\mathbf{g}_c\|_2^2}{\|\mathbf{g}_d\|_2^2} = \frac{\|\mathbf{g}_c\|_2^2 - \langle \mathbf{g}_c, \mathbf{g}_v \rangle}{\|\mathbf{g}_d\|_2^2}.$$

Note that $\|\mathbf{g}_d\|_2^2 = \|\mathbf{g}_c - \mathbf{g}_v\|_2^2 = \|\nabla_c - \nabla_v\|_2^2$ and

$$\|\mathbf{g}_c\|_2^2 - \langle \mathbf{g}_c, \mathbf{g}_v \rangle = \langle \mathbf{g}_c, \mathbf{g}_c \rangle - \langle \mathbf{g}_c, \mathbf{g}_v \rangle = \langle \mathbf{g}_c, \mathbf{g}_c - \mathbf{g}_v \rangle = \langle \nabla_c, \nabla_c - \nabla_v \rangle.$$

Thus,

$$\lambda = \frac{\langle \nabla_c, \nabla_c - \nabla_v \rangle}{\|\nabla_c - \nabla_v\|_2^2}.$$

Since the domain is restricted to $\lambda \in [0, 1]$ and $f(\lambda)$ is convex, the optimal $\lambda^*$ is the projection of the unconstrained minimizer onto $[0, 1]$:

$$\lambda^* = \max\left(0, \min\left(1, \frac{\langle \nabla_c, \nabla_c - \nabla_v \rangle}{\|\nabla_c - \nabla_v\|_2^2}\right)\right).$$

If $\mathbf{g}_d = \mathbf{0}$ (i.e., $\nabla_c = \nabla_v$), then $f(\lambda)$ is constant, and any $\lambda \in [0, 1]$ is optimal; the truncation ensures $\lambda^* \in [0, 1]$. In this case, $\lambda^* = \max\left(0, \min\left(1, \frac{\langle \nabla_c, \nabla_c - \nabla_v \rangle}{\|\nabla_c - \nabla_v\|_2^2}\right)\right)$ results in an indeterminate form, but truncation ensures that $\lambda^* \in [0, 1]$.

## C  DETAILED CONVERGENCE PROCESS

In this section, we will elaborate on the proof process of the convergence theorem. First, the following is the convergence theorem.

**Theorem 1** (Main Convergence Result). *Based on Assumptions 1-3 and Lemma 1, the model parameter $\theta_{t,e}$ trained by the proposed algorithm satisfies the following inequality after $T$ rounds of communication:*

$$\min_{t \in \{0, \ldots, T\}, e \in \{0, \ldots, \Upsilon\}} \mathbb{E}[\|\nabla \mathcal{L}(\theta_{t,e}; \lambda^*)\|^2] \leq \frac{\mathcal{L}_0 - \mathcal{L}^* + \frac{\beta\eta^2\Upsilon\sigma^2}{2}T - \tilde{\kappa}n^2 H^2 T}{\Upsilon T \left(\eta - \frac{\beta\eta^2}{2}\right)},$$

*where $\mathcal{L}_0 = \mathcal{L}(\theta_{0,0}; \lambda^*)$, $\mathcal{L}^*$ is the lower bound of the loss function, and $\tilde{\kappa} = \gamma - \frac{\beta}{2} \geq 0$. Furthermore, if the choice of the learning rate $\eta$ satisfies $\eta < \frac{2}{\beta}$ and $\tilde{\kappa} > 0$, the algorithm converges to a stationary point at a rate of $\mathcal{O}(1/T)$.*

The specific proof process is as follows. First, it is known that the local loss function $\mathcal{L}(\theta; \lambda)$ of the client is $\mathcal{L}(\theta; \lambda) = (1 - \lambda)\mathcal{L}_c(\theta) + \lambda\mathcal{L}_v(\theta)$, where $\lambda = \lambda^* = \max\left(0, \min\left(1, \frac{\langle \nabla_c, \nabla_c - \nabla_v \rangle}{\|\nabla_c - \nabla_v\|_2^2}\right)\right)$. Therefore, the gradient for a single model update is $\nabla\mathcal{L} = (1 - \lambda^*)\nabla\mathcal{L}_c + \lambda^*\nabla\mathcal{L}_v$. To clarify the derivation process, the loss function $\mathcal{L}(\theta; \lambda)$ is defined as $\mathcal{L}_{t,e}$, where $t$ represents the current communication round and $e$ represents the current local iteration round.

According to Assumption 1 ($\beta$-smoothness), the change in the loss function after parameter update is constrained. It can be derived that

$$\mathcal{L}_{t,e+1} \leq \mathcal{L}_{t,e} + \langle \nabla\mathcal{L}_{t,e}, \theta_{e+1} - \theta_e \rangle + \frac{\beta}{2}\|\theta_{e+1} - \theta_e\|^2, \tag{15}$$

where $e$ represents the number of iterations completed in the current local training, and the maximum number of iterations for local training is $\Upsilon$. Then substitute $\theta_{e+1} = \theta_e - \eta\mathbf{g}_e$, where $\mathbf{g}_e$ is a stochastic

estimate of the true gradient $\nabla \mathcal{L}_e$ calculated based on a mini-batch of data $B$. The following formula can be obtained

$$\mathcal{L}_{t,e+1} - \mathcal{L}_{t,e} \leq -\eta \langle \nabla \mathcal{L}_{t,e}, \mathbf{g}_e \rangle + \frac{\beta \eta^2}{2} \|\mathbf{g}_e\|^2. \tag{16}$$

Take the expectation of both sides of the inequality, and according to Assumption 3, we can obtain

$$\mathbb{E}[\mathcal{L}_{t,e+1} - \mathcal{L}_{t,e}] \leq -\eta \|\nabla \mathcal{L}_{t,e}\|^2 + \frac{\beta \eta^2}{2}(\sigma^2 + \|\nabla \mathcal{L}_{t,e}\|^2), \tag{17}$$

where $\eta$ is the local iterative learning rate. Its derivation process is as follows.

$$\mathbb{E}[\mathcal{L}_{t,e+1} - \mathcal{L}_{t,e}] \leq \mathbb{E}\left[-\eta \langle \nabla \mathcal{L}_{t,e}, \mathbf{g}_e \rangle + \frac{\beta \eta^2}{2} \|\mathbf{g}_e\|^2\right]$$

$$= -\eta \mathbb{E}[\langle \nabla \mathcal{L}_{t,e}, \mathbf{g}_e \rangle] + \frac{\beta \eta^2}{2} \mathbb{E}[\|\mathbf{g}_e\|^2]$$

$$= -\eta \|\nabla \mathcal{L}_{t,e}\|^2 + \frac{\beta \eta^2}{2}(\mathbb{E}[\|\mathbf{g}_e - \nabla \mathcal{L}_{t,e}\|^2] + 2\mathbb{E}[(\mathbf{g}_e - \nabla \mathcal{L}_{t,e})^\top \nabla \mathcal{L}_{t,e}] + \|\nabla \mathcal{L}_{t,e}\|^2).$$

Based on Assumption 3, where $\mathbb{E}[\mathbf{g}_e] = \nabla \mathcal{L}_{t,e}$ and $\mathbb{E}[\|\mathbf{g}_e - \nabla \mathcal{L}_{t,e}\|^2] \leq \sigma^2$, we can finally draw the conclusion that

$$\mathbb{E}[\mathcal{L}_{t,e+1} - \mathcal{L}_{t,e}] \leq -\eta \|\nabla \mathcal{L}_{t,e}\|^2 + \frac{\beta \eta^2}{2}(\mathbb{E}[\|\mathbf{g}_e - \nabla \mathcal{L}_{t,e}\|^2] + 2\mathbb{E}[(\mathbf{g}_e - \nabla \mathcal{L}_{t,e})^\top \nabla \mathcal{L}_{t,e}] + \|\nabla \mathcal{L}_{t,e}\|^2)$$

$$\leq -\eta \|\nabla \mathcal{L}_{t,e}\|^2 + \frac{\beta \eta^2}{2}(\sigma^2 + \|\nabla \mathcal{L}_{t,e}\|^2)$$

Summing up all local training losses, that is, summing up Equation equation 17, we can obtain

$$\mathbb{E}[\mathcal{L}_{t,\Upsilon} - \mathcal{L}_{t,0}] \leq \left(-\eta + \frac{\beta \eta^2}{2}\right) \sum_{e=1}^{\Upsilon} \|\nabla \mathcal{L}_{t,e}\|^2 + \frac{\beta \eta^2 \Upsilon \sigma^2}{2}. \tag{18}$$

After one layer of gradient uploading and downloading, the client model will be evaluated through linear probing in the parameter space, and finally a set of optimal combinations $(\alpha^*, \beta^*)$ will be selected to ensure Pareto improvement, that is, neither $\mathcal{L}_c$ nor $mathcalL_v$ will be reduced, and at least one of them will be improved. That is, the global update $\theta_{t+1,0} = \theta_{t,\Upsilon} - \mathbf{G}_t^{(\alpha^*,\beta^*)}$ satisfies non-degeneracy. Substituting it into the loss function before and after communication in the $t$-th round of communication, the transformation becomes,

$$\mathcal{L}_{t+1,0} - \mathcal{L}_{t,\Upsilon} \leq \langle \nabla \mathcal{L}_{t,\Upsilon}, \theta_{t+1,0} - \theta_{t,\Upsilon} \rangle + \frac{\beta}{2} \|\theta_{t+1,0} - \theta_{t,\Upsilon}\|^2$$

$$= \langle \nabla \mathcal{L}_{t,\Upsilon}, -\mathbf{G}_t^{(\alpha^*,\beta^*)} \rangle + \frac{\beta}{2} \|\mathbf{G}_t^{(\alpha^*,\beta^*)}\|^2. \tag{19}$$

The core innovation of FedMH in gradient optimization lies in the selection of $\mathbf{G}_t^{(\alpha^*,\beta^*)}$ to ensure Pareto improvement. We must formally introduce a lemma to describe the nature of this selection.

**Lemma 1** (Property of Pareto Improvement Update). *Let* $\mathbf{G}_t^{(\alpha^*,\beta^*)} = \mathbf{G}_t$, *the update direction* $\mathbf{G}_t$ *selected by the FedMH method satisfies the following conditions, where there exists a constant* $\gamma \geq \frac{\beta}{2}$:

$$\langle \nabla \mathcal{L}_{t,\Upsilon}, \mathbf{G}_t \rangle \geq \gamma \|\mathbf{G}_t\|^2$$

The proof process of Lemma 1 is as follows, It is known that the parameter update method after global communication of the client is

$$\theta_{t+1,0} = \theta_{t,\Upsilon} - \mathbf{G}_t, \tag{20}$$

where $\mathbf{G}_t$ is selected from the high-potential set $S_{\text{opt}} = S_{\text{sync}} \cup S_{\text{boundary}} \cup S_{\text{bias}}$ to satisfy the Pareto constraint while minimizing the combined loss.

$$(\alpha^*, \beta^*) = \arg\min_{\alpha,\beta} \left( \mathcal{L}_c(\theta_c - G_c^{(\alpha)}) + \mathcal{L}_v(\theta_v - G_v^{(\beta)}) \right), \tag{21}$$

It can be obtained from Equation 19 that,

$$\langle \nabla \mathcal{L}_{t,\Upsilon}, \mathbf{G}_t \rangle \leq \mathcal{L}_{t,\Upsilon} - \mathcal{L}_{t+1,0} + \frac{\beta}{2} \|\mathbf{G}_t\|^2. \tag{22}$$

According to the Pareto constraint in Section 3.3, $\mathbf{G}_t$ must satisfy,

$$\mathcal{L}_c(\theta_c - G_c^{(\alpha^*)}) \leq \mathcal{L}_c(\theta_c) + \varepsilon, \quad \mathcal{L}_v(\theta_v - G_v^{(\beta^*)}) \leq \mathcal{L}_v(\theta_v) + \varepsilon, \tag{23}$$

where $\varepsilon > 0$. This means that the update direction $\mathbf{G}_t$ is consistent with the descending directions of the two tasks, so $\langle \nabla \mathcal{L}_{t,\Upsilon}, \mathbf{G}_t \rangle > 0$, and because the advantage test prevents the rise of either component, the angle between $\nabla \mathcal{L}_{t,\Upsilon}$ and $\mathbf{G}_t$ is an acute angle.

At the same time, it is also necessary to meet the requirement of non-degradation of overall performance and ensure that the total loss decreases strictly.

$$\mathcal{L}_c(\theta_c - G_c^{(\alpha^*)}) + \mathcal{L}_v(\theta_v - G_v^{(\beta^*)}) < \mathcal{L}_c(\theta_c) + \mathcal{L}_v(\theta_v). \tag{24}$$

Since $\mathcal{L}_{t,\Upsilon}$ is the weighted sum of $\mathcal{L}_c$ and $\mathcal{L}_v$ with $\lambda^* \in [0,1]$, and the non-degenerate test implies that $\mathcal{L}_{t+1,0} - \mathcal{L}_{t,\Upsilon} < 0$ holds strictly. Substituting into equation 23, we can get,

$$\langle \nabla \mathcal{L}_{t,\Upsilon}, \mathbf{G}_t \rangle \leq \frac{\beta}{2} \|\mathbf{G}_t\|^2 - (\mathcal{L}_{t+1,0} - \mathcal{L}_{t,\Upsilon})$$
$$(\mathcal{L}_{t+1,0} - \mathcal{L}_{t,\Upsilon}) \leq \frac{\beta}{2} \|\mathbf{G}_t\|^2 - \langle \nabla \mathcal{L}_{t,\Upsilon}, \mathbf{G}_t \rangle < 0 \tag{25}$$

where since $(\mathcal{L}_{t+1,0} - \mathcal{L}_{t,\Upsilon})$ is strictly negative, we have $\frac{\beta}{2}\|\mathbf{G}_t\|^2 - \langle \nabla \mathcal{L}_{t,\Upsilon}, \mathbf{G}_t \rangle < 0$. We assume that there exists $\gamma \geq \frac{\beta}{2}$ such that,

$$\langle \nabla \mathcal{L}_{t,\Upsilon}, \mathbf{G}_t \rangle \geq \gamma \|\mathbf{G}_t\|^2$$

Substituting $\langle \nabla L_t, G_t \rangle \geq \gamma \|G_t\|^2$ into Equation 19 yields

$$\mathcal{L}_{t+1,0} - \mathcal{L}_{t,\Upsilon} \leq \langle \nabla \mathcal{L}_{t,\Upsilon}, -\mathbf{G}_t \rangle + \frac{\beta}{2} \|\mathbf{G}_t\|^2$$
$$\leq (\frac{\beta}{2} - \gamma) \|\mathbf{G}_t\|^2 \tag{26}$$
$$= -\tilde{\kappa} \|\mathbf{G}_t\|^2.$$

where $\tilde{\kappa} = \gamma - \frac{\beta}{2} \geq 0$.

Take the expectation of both sides, according to Assumption 2. Moreover, $\mathbf{G}_t$ is a linear combination of the gradients from $n$ clients, so $\|\mathbf{G}_t\| \leq \sum_{k=1}^{n} \|\mathbf{g}^{(k)}\| \leq nH$. Substituting this in, we get

$$\mathbb{E}[\mathcal{L}_{t+1,0} - \mathcal{L}_{t,\Upsilon}] \leq -\tilde{\kappa} n^2 H^2. \tag{27}$$

After a total of $T$ rounds of communication, we can obtain

$$\mathcal{L}_{0,0} - \mathcal{L}_{T,0} = \sum_{t=1}^{T} \left[ \mathcal{L}_{t,0} - \mathcal{L}_{t,\Upsilon} + \mathcal{L}_{t,\Upsilon} - \mathcal{L}_{t+1,0} \right].$$

By taking the expectation of both sides simultaneously and using the previously derived local training loss (Equation 18) and communication loss (Equation 27), we can obtain

$$\mathbb{E}[\mathcal{L}_{0,0} - \mathcal{L}_{T,0}] \geq \left( \eta - \frac{\beta \eta^2}{2} \right) \sum_{t=0}^{T} \sum_{e=0}^{\Upsilon} \mathbb{E}[\|\nabla \mathcal{L}_{t,e}\|^2] - \frac{T\beta \eta^2 \Upsilon \sigma^2}{2} + \tilde{\kappa} T n^2 H^2. \tag{28}$$

and $\mathcal{L}^* \leq \mathcal{L}_{T,0}$, therefore

$$\mathbb{E}[\mathcal{L}_{0,0} - \mathcal{L}^*] \geq \left( \eta - \frac{\beta \eta^2}{2} \right) \sum_{t=0}^{T} \sum_{e=0}^{\Upsilon} \mathbb{E}[\|\nabla \mathcal{L}_{t,e}\|^2] - \frac{T\beta \eta^2 \Upsilon \sigma^2}{2} + \tilde{\kappa} T n^2 H^2. \tag{29}$$

Therefore, we can finally obtain

$$\min_{t,e} \mathbb{E}[\|\nabla\mathcal{L}_{t,e}\|^2] \leq \frac{1}{T\Upsilon}\sum_{t=0}^{T}\sum_{e=0}^{\Upsilon}\mathbb{E}[\|\nabla\mathcal{L}_{t,e}\|^2] \leq \frac{\mathbb{E}[\mathcal{L}_{0,0}-\mathcal{L}^*] + \frac{T\beta\eta^2\Upsilon\sigma^2}{2} - \tilde{\kappa}Tn^2H^2}{T\Upsilon\left(\eta-\frac{\beta\eta^2}{2}\right)}. \quad (30)$$

This indicates that the average squared gradient norm is bounded above, which implies the method converges to a stationary point at a rate of $\mathcal{O}(1/T)$, given appropriate choices of the learning rate $\eta$ and other hyperparameters. To ensure that this inequality is meaningful in the convergence analysis, it is necessary to guarantee that the coefficient $\eta < \frac{2}{\beta}$. Otherwise, the left-hand side may not be able to constrain the sum of the gradient norms.

## D    DETAILED EXPERIMENTAL CONFIGURATION

The implementation of the baseline method uses existing frameworks (Zhang et al., 2025b;a).

### D.1    HYPERPARAMETER SETTINGS

All hyperparameter settings used in the experiment are shown in Table 5.

Table 5: Hyperparameters of the FedMH framework for federated offline handwritten signature verification.

| Hyperparameter | Symbol | Value/Range | Description |
|---|---|---|---|
| Threshold Parameter | $\vartheta$ | 0.01 | Gradient optimization minimum threshold. |
| Batch Size | $B$ | 32 | Number of samples per training iteration. |
| Learning Rate | $\eta$ | 0.01 | Step size for local gradient descent updates. |
| Communication Rounds | $T$ | 100 | Number of global communication rounds for training. |
| Dataset Split | $-$ | 30% | Proportion of data used for testing. |
| Dirichlet Parameter | $\alpha$ | 0.1 | Controls non-IID data distribution across clients. |
| Local Epochs | $\Upsilon$ | 2 | Number of local training epochs per client before aggregation. |

### D.2    DATA TRANSFORMATION SCHEME

The data transformation strategies used in the experiment refer to those in existing studies (Hamdi et al., 2021; Wang & Qi, 2023), and the selection probabilities of the corresponding data transformation methods are as follows:

- Cropping (0.4) and Rotation (0.5).
- Translation (0.4) and Brightness Adjustment (0.5).
- Contrast Adjustment (0.5) and Gamma Correction (0.4).
- Gaussian Noise Addition (0.3) and Gaussian Blur (0.3).
- Clipping (0.3) and Motion Blur (0.2).

### D.3    MODEL STRUCTURE DESCRIPTION

FedMH adopts a dual-head architecture, which includes a fully connected classification head for author identification and a binary classification head for forgery detection. Both operate based on the shared feature representations extracted by the common backbone network. The optimization process uses Stochastic Gradient Descent (SGD) to jointly minimize a weighted combination of the cross-entropy loss for author classification (calculated only based on real signatures) and the log-based binary cross-entropy loss for forgery detection (applied to all samples). When only forged samples exist in a batch, the system will adaptively adjust $\lambda$ to 1.0, thereby maintaining stable training dynamics throughout the federated learning process. The settings of other models are shown in Table 6 and Table 7.

Table 6: Architecture specifications for signature verification models. All models accept single-channel input images and output $d$-dimensional feature representations.

| Model | Base Architecture | Feature Dim ($d$) | Patch Size | Pre-training |
|---|---|---|---|---|
| ViTSigNet | ViT-B/16 | 768 | $16 \times 16$ | ImageNet-21k |
| ViTSigNet-32 | ViT-B/32 | 768 | $32 \times 32$ | ImageNet-21k |
| SigNet | CNN (AlexNet-like) | 2048 | N/A | None |
| SigNet-S | CNN (Reduced) | 2048 | N/A | None |
| OctConvNet | OctaveConv | 2048 | N/A | None |

Table 7: Layer-wise architecture details. $\mathrm{Conv}(c_{\mathrm{in}}, c_{\mathrm{out}}, k, s)$ denotes convolution with $c_{\mathrm{in}}$ input channels, $c_{\mathrm{out}}$ output channels, kernel size $k$, and stride $s$. $\mathrm{OctConv}(c, \alpha)$ denotes octave convolution with $c$ channels and ratio $\alpha$.

| Model | Layer Specification |
|---|---|
| ViTSigNet | $\mathrm{Conv2d}(1, 768, 16, 16) \rightarrow$ Transformer Blocks $\times 12$ $\rightarrow \mathrm{Linear}(1000, 6400) \rightarrow \mathrm{BN} \rightarrow \mathrm{ReLU}$ $\rightarrow \mathrm{Linear}(6400, 2048) \rightarrow \mathrm{BN} \rightarrow \mathrm{ReLU} \rightarrow \mathrm{Linear}(2048, C)$ |
| ViTSigNet-32 | $\mathrm{Conv2d}(1, 768, 32, 32) \rightarrow$ Transformer Blocks $\times 12$ $\rightarrow \mathrm{Linear}(1000, 6400) \rightarrow \mathrm{BN} \rightarrow \mathrm{ReLU}$ $\rightarrow \mathrm{Linear}(6400, 2048) \rightarrow \mathrm{BN} \rightarrow \mathrm{ReLU} \rightarrow \mathrm{Linear}(2048, C)$ |
| SigNet | $\mathrm{Conv}(1, 96, 11, 4) \rightarrow \mathrm{MaxPool}(3, 2) \rightarrow \mathrm{Conv}(96, 256, 5, 1)$ $\rightarrow \mathrm{MaxPool}(3, 2) \rightarrow \mathrm{Conv}(256, 384, 3, 1) \rightarrow \mathrm{Conv}(384, 384, 3, 1)$ $\rightarrow \mathrm{Conv}(384, 256, 3, 1) \rightarrow \mathrm{MaxPool}(3, 2)$ $\rightarrow \mathrm{FC}(6400, 2048) \rightarrow \mathrm{FC}(2048, 2048) \rightarrow \mathrm{FC}(2048, C)$ |
| SigNet-S | $\mathrm{Conv}(1, 96, 11, 4) \rightarrow \mathrm{MaxPool}(3, 2) \rightarrow \mathrm{Conv}(96, 256, 5, 1)$ $\rightarrow \mathrm{MaxPool}(3, 2) \rightarrow \mathrm{Conv}(256, 384, 3, 1) \rightarrow \mathrm{Conv}(384, 256, 3, 1)$ $\rightarrow \mathrm{MaxPool}(3, 2) \rightarrow \mathrm{FC}(6400, 2048) \rightarrow \mathrm{FC}(2048, C)$ |
| OctConvNet | $\mathrm{OctConv}(1 \rightarrow 16, \alpha_{\mathrm{out}} = 0.5) \rightarrow \mathrm{MaxPool}(2, 2)$ $\rightarrow \mathrm{OctConv}(16 \rightarrow 8, \alpha = 0.5) \rightarrow \mathrm{MaxPool}(2, 2)$ $\rightarrow \mathrm{OctConv}(8 \rightarrow 4, \alpha_{\mathrm{in}} = 0.5, \alpha_{\mathrm{out}} = 0)$ $\rightarrow \mathrm{FC}(12544, 6400) \rightarrow \mathrm{FC}(6400, 2048) \rightarrow \mathrm{FC}(2048, C)$ |

## D.4 EXPERIMENTAL CONFIGURATION

**Dataset:** Experiments are conducted on the **GPDS-10000, Bengali, and Hindi datasets** to evaluate the performance of the proposed FedMH. The GPDS-10000 dataset is a large benchmark dataset, in which 24 genuine signatures and 30 skilled forged signatures from each of 10,000 users are included, with a total of 240,000 genuine samples and 300,000 forged samples. Both the Bengali and Hindi subsets are derived from the BHSig260 corpus; 100 users are included in the former, and 160 users are included in the latter.

**Performance Metrics:** To evaluate the verification performance of the proposed method, the following metrics are taken into account. **AUC**: Area under the ROC curve. **FRR**: the rate at which genuine signatures are incorrectly classified as forgeries. **FAR**: the ratio of skilled forgeries that are incorrectly validated as genuine signatures. **Equal Error Rate (EER)**: the error when FAR = FRR.

**Training settings:** The client locally sets the batch size to 32 and the local learning rate to 0.01, with a total of 100 rounds of global training, and 2 local epochs per round. The total number of participating clients is set to two configurations: 5 clients and 20 clients. The feature dimension extracted by the final feature extractor is 2048. The experiment uses the first 1000 users from the GPDS dataset, and the training process only uses the last 700 users among these 1000 users. The signature samples of these 700 users are divided into clients according to the Dirichlet distribution with the parameter $\alpha = 0.1$ to simulate the heterogeneous distribution in real-world scenarios. Meanwhile, to demonstrate the generality of FedMH, we use two client model settings: one is the client **homogeneous** model setting, and the other is the client **heterogeneous** model setting. The models involved include **SigNet**, **OctConvNet**, and **ViTs**. The heterogeneous model is set as follows

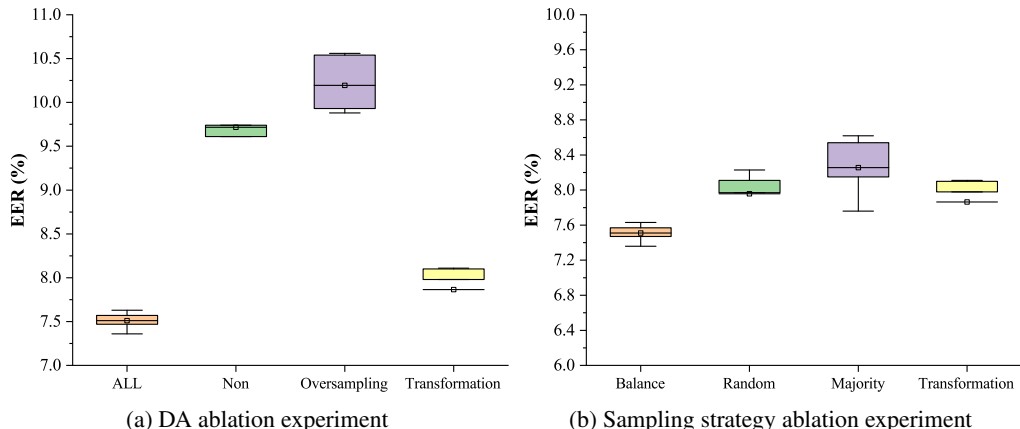

(a) DA ablation experiment

(b) Sampling strategy ablation experiment

Figure 4: Experimental results of ablation studies on the DA module components using a single client and the GPDS dataset. (a) The ablation results of data transformation strategies and oversampling strategies in the DA module. **ALL** (Complete Strategy), **Non** (No Enhancement), **Oversampling** (Oversampling Only), and **Transformation** (Transformation Only). (b) The ablation results using the same data transformation strategy under different sampling strategies. **Balance** (for minority sample augmentation), **Random** (random sampling), **Majority** (for majority sample augmentation) and **Transformation** (Transformation Only). All sampling strategies in the figure result in the same final data volume.

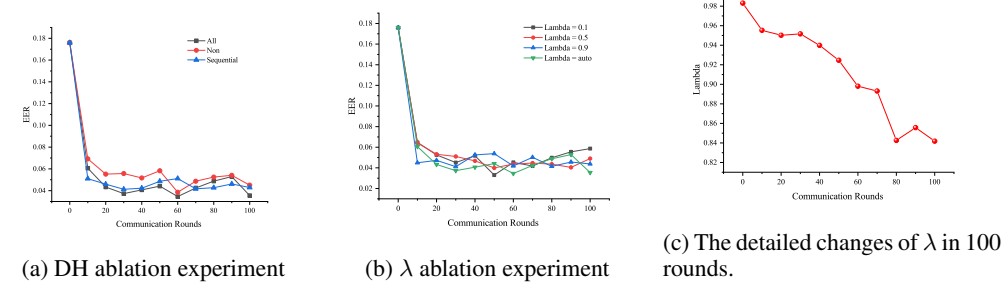

(a) DH ablation experiment     (b) $\lambda$ ablation experiment     (c) The detailed changes of $\lambda$ in 100 rounds.

Figure 5: Results of the ablation experiment on the DH module with a single client and the GPDS dataset.

**HSigNet uses SigNet and SigNet_S**, **HOctConvNet uses OctConvNet and SigNet**, while **HViTs use ViTSigNet and ViTSigNet-32**. The improvement threshold $\vartheta$ is set to meet a $1\%$ increase.

**Test settings:** We use a support vector machine in a Writer-Dependent(WD) environment to evaluate the quality of features extracted by the client model's feature extractor. The specific settings are as follows: in the generalization experiment, an SVM is used as the classifier for five-fold cross-validation. The input image size is $224 \times 224$ pixels. Each user uses 12 real signatures for training, 10 real signatures for testing, and 12 forged signatures are selected as negative samples. The SVM classifier uses the RBF kernel by default, with a regularization parameter $C$ of 1 and gamma of $2^{-11}$. The batch size is 32. In the personalization experiment, we used 700 users from the GPDS dataset to participate in model training, and $30\%$ of the data was used as the test set to verify the personalization performance of the model, where the users used for training and testing of each model are the same. Except for the generalization experiment, where the metric is the average performance of client models **after 100 training rounds**, all other experiments use the average model performance over **the last 10 rounds of 100 communication rounds** as the result metric.

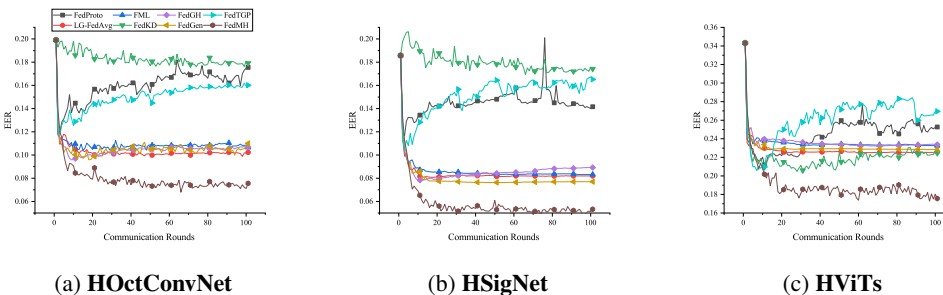

(a) **HOctConvNet**          (b) **HSigNet**          (c) **HViTs**

Figure 6: The training process under the GPDS-700 dataset and heterogeneous model settings, where the performance changes are shown through the **average EER** of each method across 5 clients.

# E  OTHER EXPERIMENTAL RESULTS

Figure 4 provides an in-depth analysis of the effectiveness of the DA module. From Figure 4(a), we can see that among the four DA module configurations: **ALL** (complete strategy), **Non** (no augmentation), **Oversampling** (oversampling only), and **Transformation** (transformation only), the **ALL** configuration has the lowest EER and the most concentrated distribution. It should also be noted that the **Oversampling** configuration has the worst EER performance and a very scattered data distribution, indicating that its performance is the poorest and most unstable. Therefore, using the oversampling strategy alone will lead to overfitting of the model and a decrease in generalization performance due to duplicate data. However, combining data transformation and oversampling achieves better results than using data transformation alone, which shows that the DA module suppresses the overfitting phenomenon of the model by using these two strategies.

From Figure 4(b), we can see that the **Balance** sampling strategy (for enhancing minority samples) used in the DA module achieves the best performance, while the **Random** (random sampling) and **Majority** (for enhancing majority samples) sampling strategies fail to effectively exert the data augmentation effect of data transformation, and their performance is even worse compared to **Transformation** (using data transformation directly without resampling). This indicates that the sampling strategy of the DA module is also necessary, and data transformation cannot play an overwhelming role.

From the different results of the three different settings in Figure 5(a): **All** using the complete DH module, **Sequential** performing forgery detection before user identification, and **Non** using a single-task head, we can see that the complete DH shows better performance than the other two settings both in the early stage of model training (20-60 rounds) and the late stage of training (90-100 rounds). It is worth noting that the **Sequential** setting begins to overfit from round 70, which indicates that the sequential design method will reduce the model performance in the later stage due to gradient conflicts. From Figure 5(b) and Figure 5(c), we can see that the dynamic $\lambda$ weights of the DH module can provide more flexible weight selection during the commonly used two-headed training for signatures, and further optimization is performed when other static weight settings show overfitting in the later stages of training (90-100 rounds).

Figure 6 details the training dynamics over 100 communication rounds under the GPDS-700 heterogeneous setup. This figure intuitively reveals the vulnerability of standard heterogeneous federated learning methods when facing data scarcity and task conflicts. Multiple baseline methods exhibit severe performance degradation in the middle and later stages of training. Their EER increases instead of decreasing after the initial communication rounds (approximately 10-20 rounds), showing a significant negative optimization phenomenon, indicating that continuous communication leads to catastrophic forgetting or conflicts of model knowledge. In sharp contrast, FedMH (black curve) can consistently maintain a low EER level after rapid convergence and effectively suppresses such negative optimization and overfitting. This is mainly attributed to our client-adaptive data augmentation strategy (DA), which mitigates the data imbalance problem locally, providing more robust gradients for the model, thereby enabling stable multi-task aggregation in heterogeneous environments.

Table 8: Under the SigNet model setup, the communication cost per iteration after 5 clients (M=5) participate in training. $\Theta$ represents the parameters of the auxiliary generator in FedGen. $|\theta_g|$ and $|w_g|$ denote the parameters of the auxiliary feature extractor and classifier in FML and FedKD, respectively. $r$ is the compression rate parameter controlled by SVD parameter decomposition in FedKD. $|\theta_g| \gg K \times C$, where $K$ is the feature dimension and $C$ is the total number of global categories. $C_i$ represents the number of categories on client $i$. For FedMH, $|g_i^c|$ and $|g_i^f|$ represent the gradients of the user classification head and the authenticity verification head, respectively. $|G|$ is the cumulative gradient set.

| Methods | Theory | Practice |
|---------|--------|----------|
| LG-FedAvg | $\sum_{i=1}^{M} \|w_i\| \times 2$ | 10.01 MB |
| FedGen | $\sum_{i=1}^{M}(\|w_i\| \times 2 + \|\Theta\|)$ | 35.10 MB |
| FML | $M \times (\|\theta_g\| + \|w_g\|) \times 2$ | 867.36 MB |
| FedGH | $\sum_{i=1}^{M} K \times C_i + M \times \|w_g\|$ | 5.09 MB |
| FedKD | $M \times (\|\theta_g\| + \|w_g\|) \times 2 \times r$ | 30.20 MB |
| FedProto | $\sum_{i=1}^{M} K \times (C_i + C)$ | 0.16 MB |
| FedTGP | $\sum_{i=1}^{M} K \times (C_i + C)$ | 0.16 MB |
| FedMH | $\sum_{i=1}^{M}(\|g_i^c\| + \|g_i^f\|) + \|G\|$ | 194.17 MB |

Table 9: Statistical results of the single-round communication and computation costs after 5 clients participate under the HSigNet setup and the GPDS-700 dataset. Among them, **FLOPs** count the number of GPU floating-point operations.

| Methods | FLOPs |
|---------|-------|
| LG-FedAvg | 91.48 TFLOPs |
| FedGen | 92.00 TFLOPs |
| FML | 198.64 TFLOPs |
| FedGH | 135.54 TFLOPs |
| FedKD | 201.29 TFLOPs |
| FedProto | 91.48 TFLOPs |
| FedTGP | 135.55 TFLOPs |
| FedMH | 327.53 TFLOPs |

The communication theory for the baselines in Table 8 refers to existing summaries, as shown in Table 8. The communication cost of FedMH (194.17 MB) is higher than that of most baselines. This is an expected result of our architecture because the Dual-task Head (DH) mechanism needs to convey gradient information for both classification and validation tasks simultaneously, which is different from single-task methods. We accept the higher communication overhead in exchange for significant accuracy improvements in heterogeneous and multi-task scenarios, as shown in Table 2 and Figure 2.

Table 9 shows the computational cost. We must acknowledge that due to the local execution of the DH and GO modules on the client side, the TFLOPs of FedMH (327.53 TFLOPs) is significantly higher than that of the baseline. However, this higher computational cost directly translates into the superior performance and robustness of our method.

It can also be seen from Figure 7 that FedMH has a relatively high improvement compared to other heterogeneous federated learning methods in the case of multiple clients.

The experiment followed the ablation experiment setup. As can be seen from Table 10, the effect of gradient optimization using $S_{\textbf{all}}$ is worse than that using $S_{\textbf{opt}}$, and the time complexity of $S_{\textbf{all}}$ is much higher than that of $S_{\textbf{opt}}$. This confirms that under the setting where optimization stops when the growth meets the improvement threshold $\vartheta$, $S_{\textbf{opt}}$ prioritizes testing the gradient binary combinations that are most likely to achieve the best optimization.

Figure 8 shows the robustness of the FedMH method to key hyperparameters on the HSigNet model and the GPDS-700 dataset. The results are obtained by comparing different batch sizes, local train-

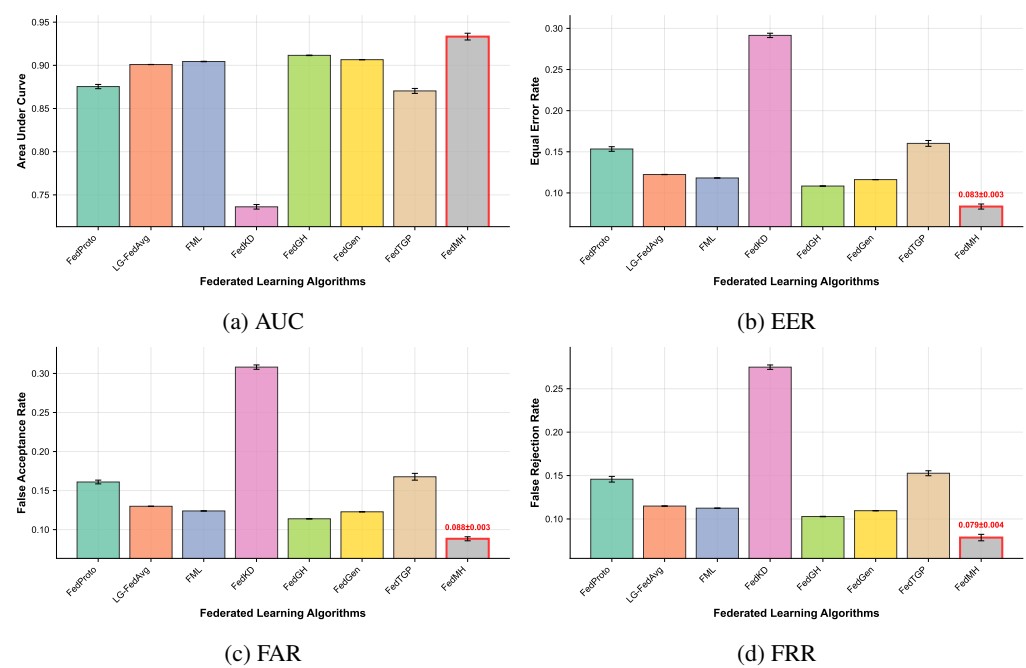

Figure 7: The training process under the GPDS-700 dataset and the HSigNet model, which involves the average performance of the last 10 rounds of models from 20 clients using various methods

Table 10: Performance under different gradient binary combination settings and the GPDS-700 dataset, where $S_{\mathbf{all}}$ represents all non-repetitive gradient binary combinations.

| Gradient Set | | Time Complexity | Performance indicators (%) | | | |
|---|---|---|---|---|---|---|
| $S_{\mathbf{opt}}$ | $S_{\mathbf{all}}$ | | EER $\downarrow$ | AUC $\uparrow$ | FAR $\downarrow$ | FRR $\downarrow$ |
| ✗ | ✗ | $O(n)$ | $5.54 \pm 0.25$ | $96.70 \pm 0.16$ | $5.74 \pm 0.25$ | $5.33 \pm 0.24$ |
| ✓ | ✗ | $O(n)$ | $\mathbf{5.28 \pm 0.17}$ | $\mathbf{96.91 \pm 0.13}$ | $\mathbf{5.47 \pm 0.18}$ | $\mathbf{5.09 \pm 0.17}$ |
| ✗ | ✓ | $O(n^2)$ | $5.78 \pm 0.12$ | $96.56 \pm 0.10$ | $6.00 \pm 0.16$ | $5.56 \pm 0.11$ |

ing epochs, and the significance threshold $\vartheta$ in Pareto gradient optimization, respectively. It is shown that the convergence curves and final performance of EER remain highly consistent and stable under different settings of local training epochs and significance threshold $\vartheta$. Although there are differences in the final EER performance under different batch size settings, the differences are small. This indicates that the FedMH method is basically insensitive to the selection of these hyperparameters, verifying the rationality of the experimental configuration and the stability of the method.

Table 11: The results of the ablation experiments using the HSigNet model and the GPDS-300 dataset, where the performance metrics are the average of the performance metrics of the 5 client models in 5-fold cross-validation. Here, **DA** refers to the data augmentation module, **DH** is the dual-task head mechanism, and **GO** refers to the gradient optimization module.

| Module Configuration | | | Performance (EER %) |
|---|---|---|---|
| DA | DH | GO | |
| ✗ | ✗ | ✗ | $9.96 \pm 0.46$ |
| ✓ | ✗ | ✗ | $8.05 \pm 0.35$ |
| ✓ | ✓ | ✗ | $7.52 \pm 0.55$ |
| ✓ | ✓ | ✓ | $\mathbf{7.43 \pm 0.62}$ |

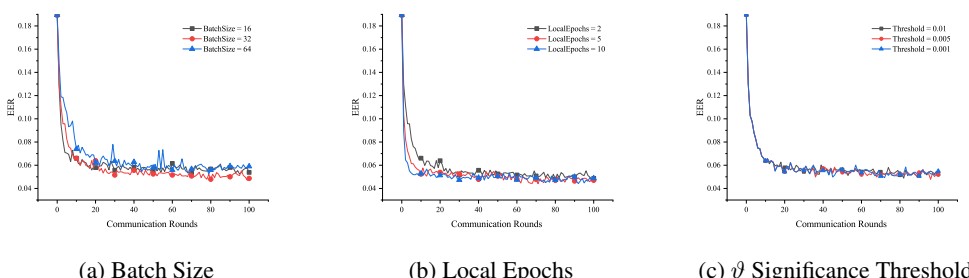

(a) Batch Size       (b) Local Epochs       (c) $\vartheta$ Significance Threshold

Figure 8: Results of hyperparameter ablation experiments on the HSigNet model setup and GPDS-700 dataset

Table 12: The TFLOPs results of different clients of the GO module under settings with different numbers of clients. The calculation cost is related to the number of clients $n$ and the Pareto index $(i, j)$ dynamically selected during the search process.

| Clients ($n$) | Number of Searches | Reported TFLOPs |
|---|---|---|
| $n = 2$ | min | 38.58 TFLOPs |
| | max | 96.44 TFLOPs |
| $n = 4$ | min | 136.47 TFLOPs |
| | max | 136.47 TFLOPs |
| $n = 5$ | min | 38.34 TFLOPs |
| | max | 151.63 TFLOPs |

Table 12 confirms that the computational overhead of the GO module is highly dynamic and related to two key factors. First, the maximum TFLOPs increase with the number of clients $n$, rising from 96.44 TFLOPs when $n = 2$ to 151.63 TFLOPs when $n = 5$, which empirically supports our $O(n)$ complexity analysis. Second, the overhead is not fixed in each round, which is clearly reflected in the significant gap between the minimum (38.34 TFLOPs) and maximum (151.63 TFLOPs) costs when $n = 5$. This fluctuation is directly attributed to the number of searches for the specific indices $(i, j)$ selected by the Pareto search, demonstrating that the computational cost is a necessary data-driven expenditure incurred to achieve optimal aggregation, which is fully consistent with the complexity of the strategy selected in each round.

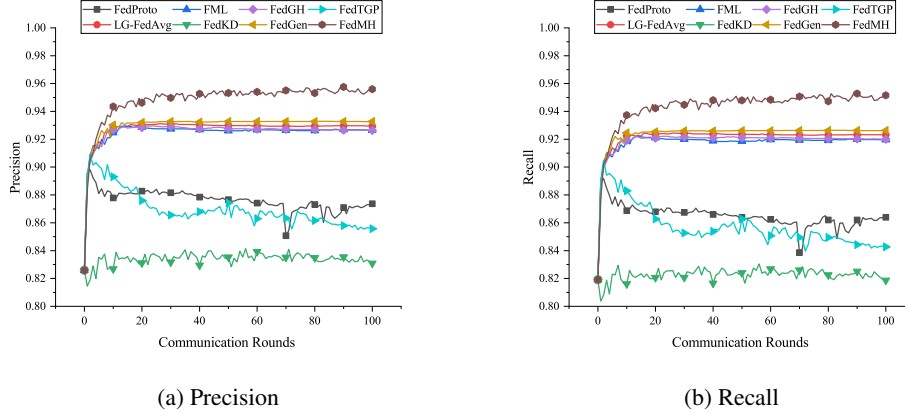

(a) Precision             (b) Recall

Figure 9: The precision and recall results of different federated learning algorithms under the HSigNet model setup and GPDS-700 dataset.

Table 13: The enhancement effect of the DA module on federated learning methods under the HSigNet model settings and the GPDS-700 dataset.

| Method | Use DA | | | | Not Use DA | | | |
|---|---|---|---|---|---|---|---|---|
| | EER | AUC | FAR | FRR | EER | AUC | FAR | FRR |
| FedProto | 9.44 | 93.67 | 9.64 | 9.24 | 13.6 | 90.8 | 13.9 | 13.3 |
| FedKD | 13.18 | 90.18 | 13.74 | 12.62 | 18.4 | 85.8 | 19.2 | 17.6 |
| FedTGP | 9.04 | 94.53 | 9.23 | 8.84 | 16.1 | 88.3 | 16.4 | 15.7 |

Figure 9 shows the variation trajectories of Precision and Recall of different federated learning algorithms over 100 communication rounds under the settings of the HSigNet heterogeneous model and the GPDS-700 dataset. The experimental results indicate that the proposed FedMH method achieves rapid convergence in the early stages of training and maintains significantly better performance stability than the comparative baselines throughout the communication process. It overcomes the performance fluctuations and negative optimization phenomena that occur in other heterogeneous federated learning methods, such as LG-FedAvg and FedProto, in the middle and late stages of training due to gradient conflicts or catastrophic forgetting caused by model heterogeneity. Notably, the Precision and Recall curves of FedMH show a high degree of consistency, which is attributed to the experiment following the dynamic threshold selection strategy based on the Equal Error Rate (EER) criterion in the scenario of handwritten signature verification (Hafemann et al., 2017). Specifically, on an approximately balanced test set containing 10 genuine signatures and 12 forged signatures, the False Acceptance Rate (FAR) and False Rejection Rate (FRR) are forced to be balanced. This consistency strongly demonstrates that FedMH, through the dual-task head mechanism and Pareto gradient optimization, successfully establishes a robust balance between user classification and authenticity verification tasks, thereby achieving unbiased discriminative ability that can accurately identify genuine signatures without missing positive samples.

As shown in the Table 13, the DA module has universality in data heterogeneous scenarios. However, in terms of the performance of FedKD that uses the DA module, it is still not as good as most methods in Table 2. This indicates that the DA module alone cannot solve the overfitting phenomenon caused by errors in communication strategies. Furthermore, it reflects the necessity of the DH and GO modules in the proposed FedMH framework.

## F THE APPLICATION OF PARETO OPTIMIZATION IN THE FEDMH FRAMEWORK

### F.1 COMPARISON WITH CURRENT MULTI-OBJECTIVE FEDERATED LEARNING RESEARCH

In the field of multi-objective federated learning (MOFL), related works use different optimization algorithms to address the balance between multiple conflicting objectives. For example, the FedMGDA+ algorithm is introduced to demonstrate that this method can converge to a Pareto stationary solution (Hu et al., 2022). The FedVal system estimates the impact of adversarial or non-cooperative clients on model unfairness by establishing a central validation set to assign validation scores (Mehrabi et al., 2022). The CMOFB algorithm, targeting the SecureBoost framework, minimizes utility loss, training costs, and privacy leakage simultaneously by searching for optimal parameters (Ren et al., 2023). Among them, the proposed FedMH framework has similarities with the FedMGDA+ algorithm and the CMOFB algorithm. For example, they all use Pareto stationary solutions to aid convergence and search for optimal parameters to minimize losses simultaneously.

However, traditional MOFL paradigms, such as FedMGDA+, typically adopt a server-centric perspective, relying on the server to collect all gradients and solve complex quadratic programming problems to compute the common descent direction (Skovajsova et al., 2025). In contrast, the proposed FedMH innovatively delegates optimization decisions to the clients. Instead of pursuing an accurate analytical solution in the continuous space, it proposes a discrete search protocol based on a predefined high-potential set ($S_{opt}$). This mechanism allows clients to perform linear probing in their local parameter space and independently find the gradient combination step size that maximizes the performance of their local models. It not only reduces the time complexity from $O(n^2)$ to

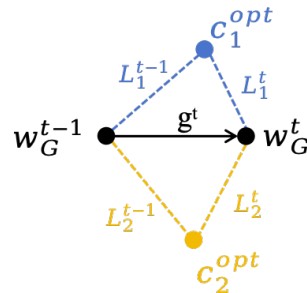

$c_n^{opt}$: Optimal parameters for Client $\boldsymbol{n}$.

$w_G^T$: Global model parameters at the $\boldsymbol{T}$-th global round.

$g^T$: Gradient of the $\boldsymbol{T}$-th round of global model update.

$L_n^T$ : The loss between the global parameters and the optimal parameters of client $\boldsymbol{n}$ in the $\boldsymbol{T}$-th global round.

Figure 10: Understand Pareto optimization from the perspective of global model parameter updates, taking two clients as an example.

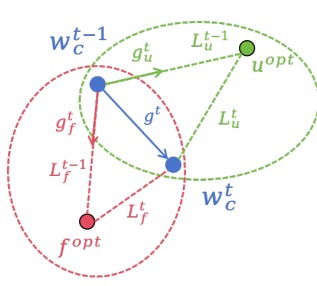

$w_c^T$ : Parameter of the client at the $\boldsymbol{T}$-th round.

$g^T$ : The composite gradient updated by the client after the $\boldsymbol{T}$-th round of global communication.

$g_u^T$ : The gradient of the user classification task updated by the client after the $\boldsymbol{T}$-th round of global communication.

$g_f^T$ : The gradient of the authenticity identification task updated by the client after the $\boldsymbol{T}$-th round of global communication.

$L_u^T$ : The loss between the client and the optimal parameters of the user classification task during the $\boldsymbol{T}$-th round of global communication updates.

$L_f^T$ : The loss between the client and the optimal parameters of the authenticity identificationtask during the $\boldsymbol{T}$-th round of global communication updates.

Figure 11: Under the proposed FedMH framework, to meet the requirements of offline handwritten signature verification, the model updates are shifted from the server to the client side.

$O(n)$ but also enables the effective application of multi-objective optimization theory in federated scenarios with heterogeneous models.

Moreover, traditional MOFL methods are mainly committed to resolving inter-client objective conflicts, such as fairness or heterogeneous objectives. In contrast, FedMH, targeting the multi-task requirements of handwritten signature verification, takes into account the intra-client application of multi-objective optimization. It specifically addresses the concurrent dual-task conflicts on a single node, filling the gap in federated learning in handling multi-task conflicts on a single node.

### F.2 WHY USE PARETO UPDATE

As shown in Figure 10, from the perspective of global model parameter updates, Pareto update can simultaneously take into account the optimal parameter points of multiple clients, and the details of the update are shown in Equation equation 31. This ensures that the global model optimizes its parameters without compromising the performance of any participant, thereby naturally ensuring fairness among users, providing participation incentives, and effectively defending against poisoning attacks by malicious users.

$$\begin{cases} L_1^{t-1} \geq L_1^t, \\ L_2^{t-1} \geq L_2^t, \end{cases} \tag{31}$$

where $L_n^T$ represents the loss between the global parameters and the optimal parameters of client $n$ in the $T$-th global round. This formula requires that updating one objective must not harm other objectives.

As shown in Figure 11, aiming at the requirements of the offline handwritten signature verification scenario and the problem of data heterogeneity existing in different nodes, the proposed FedMH framework places the Pareto update on the client side. It can be seen from Figure 6 that under this setting, the proposed FedMH framework can indeed solve the problems of data heterogeneity and multi-objective optimization among different clients.

## G    INSTRUCTIONS ON THE USAGE OF LLMS

In the process of writing this article, LLMs are used for grammar correction and improvement. And use LLMs to check if there are any errors in the paper.

