# OpenReview forum: "FedMH: Federated Learning with Multi-Task Head for Heterogeneous Models in Offline Signature Verification"
_ICLR.cc/2026/Conference — ICLR 2026 Conference Desk Rejected Submission_

### Official Review · Reviewer_9tRX · 2025-10-29

**Soundness:** 3
**Presentation:** 2
**Contribution:** 2
**Rating:** 4
**Confidence:** 4

**Summary:**

**FedMH** proposes a novel **heterogeneous federated learning framework** for **offline handwritten signature verification**, aiming to achieve high-accuracy authentication under privacy constraints, data imbalance, and task heterogeneity. The method introduces a **multi-task head collaboration mechanism** that jointly optimizes user identification and forgery detection, enabling effective knowledge sharing and generalization across heterogeneous model architectures.

**Strengths:**

1. The paper introduces a Pareto-based gradient optimization framework that provides a theoretically grounded means of balancing competing objectives in multi-task federated learning. This design effectively mitigates gradient conflicts and enhances optimization stability across heterogeneous clients.
2. The authors present a relatively comprehensive non-convex convergence proof, establishing theoretical reliability under federated and heterogeneous settings. Such formal analysis is rarely provided in applied FL tasks and enhances the paper’s scientific rigor.

**Weaknesses:**

1. For each client k, data from a specific user a are divided into *genuine* and *forged* signatures. As mentioned around line 170, genuine samples typically far outnumber forged ones. In the following paragraph, the minority-class enhancement is implemented via **oversampling** until the sample count matches that of the majority class. This approach may lead to **severe overfitting of the minority class**, particularly detrimental to the generalization of the downstream **forgery-detection** task.
2. According to the authors, the number of samples may vary substantially across users (e.g., user a vs. user b). It is unclear whether this imbalance across users was explicitly considered during the data-augmentation process.
3. The proposed data-augmentation module appears to rely on **standard transformations**—as shown in Appendix A, the augmentation functions are mostly existing operations combined into a random transformation sequence. According to the ablation results (Table 3), the DA module contributes the most to the overall performance, whereas the other two modules show **marginal improvements**.
4. It is unclear whether the **baseline methods** in the main experiments were trained with the **same data-augmentation setup** for fair comparison.
5. The experiments could be made more convincing by testing with a **larger number of clients** (e.g., 20 or 50) to evaluate scalability and robustness.
6. The authors may consider discussing **related approaches that perform forgery detection prior to user identification**, and clarifying how the proposed **dual-task mechanism** offers advantages over such sequential designs.

**Questions:**

I may raise the score if the authors clarify/resolve the above concerns 1, 2, 3, 4, 5, and 6.

---

> ### Author Response · Authors · 2025-11-17
> **Instructions and Responses for Paper Revision**
>
> Dear Reviewer,
>
> Thank you for reviewing our paper and providing your suggestions. Below, we will address your questions and explain the changes made to improve the paper.
>
> ---
>
> Q1：
>
> For each client k, data from a specific user a are divided into genuine and forged signatures. As mentioned around line 170, genuine samples typically far outnumber forged ones. In the following paragraph, the minority-class enhancement is implemented via oversampling until the sample count matches that of the majority class. This approach may lead to severe overfitting of the minority class, particularly detrimental to the generalization of the downstream forgery-detection task.
>
> A1：
>
> We conducted ablation experiments on your question, and the results are shown in Figure 4(a). Using the oversampling method alone does lead to overfitting. However, when combined with traditional enhancement techniques, it can achieve an effect where 1+1>2. This indicates that the oversampling enhancement method can effectively improve the performance of traditional enhancement techniques.
>
> ---
>
> Q2：
>
> According to the authors, the number of samples may vary substantially across users (e.g., user a vs. user b). It is unclear whether this imbalance across users was explicitly considered during the data-augmentation process.
>
> A2：
>
> The DA method mainly addresses the class imbalance phenomenon between genuine and forged signatures within a user, but in future work, we will further consider the imbalance phenomenon between users.
>
> ---
>
> Q3：
>
> The proposed data-augmentation module appears to rely on standard transformations—as shown in Appendix A, the augmentation functions are mostly existing operations combined into a random transformation sequence. According to the ablation results (Table 3), the DA module contributes the most to the overall performance, whereas the other two modules show marginal improvements.
>
> A3：
>
> Based on your question, we have added an analysis of the ablation experiments. As shown in Table 11, in the heterogeneous scenario, the marginal contributions of the DH and GO modules increase significantly. The DA module alone reduces the EER from $9.96 \pm 0.46$ to $8.05 \pm 0.35$. After adding the DH and GO modules, the EER is further reduced to $7.43 \pm 0.62$, resulting in an additional EER reduction of 0.62%. Compared with the marginal gain of 0.3% in the homogeneous setting (5.58 $\rightarrow$ 5.28), the gain in the heterogeneous setting (0.62%) is more than twice as much.
>
> Table 11： The results of the ablation experiments using the HSigNet model and the GPDS-300 dataset, where the performance metrics are the average of the performance metrics of the 5 client models in 5-fold cross-validation. Here, \textbf{DA} refers to the data augmentation module, \textbf{DH} is the dual-task head mechanism, and \textbf{GO} refers to the gradient optimization module.
> | DA | DH | GO | Performance (EER %) |
> |----|----|----|----------------------|
> | ×  | ×  | ×  | \$9.96 \pm 0.46\$    |
> | √  | ×  | ×  | \$8.05 \pm 0.35\$    |
> | √  | √  | ×  | \$7.52 \pm 0.55\$    |
> | √  | √  | √  | \$7.43 \pm 0.62\$    |
>
> ---
>
> Q4：
>
> It is unclear whether the baseline methods in the main experiments were trained with the same data-augmentation setup for fair comparison.
>
> A4：
>
> Before training, we applied the same traditional data transformations to the training data of the baseline methods. However, since DA is a module of our proposed FedMH method, the data augmentation strategy based on oversampling was only used for the FedMH method, while all other data processing strategies remained consistent.
>
> ---
>
> Q5：
>
> The experiments could be made more convincing by testing with a larger number of clients (e.g., 20 or 50) to evaluate scalability and robustness.
>
> A5：
>
> From Figure 7, we can see the performance comparison between FedMH and other baseline methods under 20 clients. FedMH also has a relatively high improvement compared with other heterogeneous federated learning methods in the case of multiple clients.
>
> ---
>
> Q6：
>
> The authors may consider discussing related approaches that perform forgery detection prior to user identification, and clarifying how the proposed dual-task mechanism offers advantages over such sequential designs.
>
> A6：
>
> Thank you for your suggestion. We have added relevant ablation experiments. As can be seen from Figure 5, the proposed dual-task mechanism has advantages. In the sequential design, the classification head trained later will affect the results of the classification head trained earlier. The complete DH shows better performance than the other two settings both in the early stage of model training (20-60 rounds) and the late stage of training (90-100 rounds). It is worth noting that the \textbf{Sequential} setting starts to overfit from round 70, which indicates that the sequential design method will reduce the model performance in the later stage due to gradient conflicts.
>
> ---
> Thank you for your suggestion.

---

> > ### Author Response · Authors · 2025-11-21
> > **Supplementary to Question 3**
> >
> > Dear Reviewer,
> >
> > In response to your question, we have supplemented more detailed experiments on the DA module, and the results are shown in Figure 4(b). Although data transformation does provide significant help to the DA module, it can be seen from the figure that the sampling strategy in the DA module plays a positive role under the condition of the same number of samples. However, the \textbf{Random} (random sampling) and \textbf{Majority} (enhancement for the majority of samples) sampling strategies cannot effectively exert the data augmentation effect of data transformation, and their performance is worse compared with \textbf{Transformation} (using data transformation directly without resampling). This indicates that the sampling strategy of the DA module is also necessary, and data transformation cannot play an overwhelming role.

---

### Official Review · Reviewer_Gxsr · 2025-10-30

**Soundness:** 2
**Presentation:** 3
**Contribution:** 2
**Rating:** 4
**Confidence:** 5

**Summary:**

The paper proposes FedMH, a novel heterogeneous federated learning framework tailored for offline handwritten signature verification, addressing three major challenges in this domain: (1) local data scarcity and severe class imbalance between genuine and forged signatures; (2) the necessity of handling two interdependent but distinct tasks—user classification and forgery detection; (3) the difficulty of aggregating heterogeneous models in FL environments. The main contributions of FedMH include: (1) a  Client-Adaptive Data Augmentation (DA) technique; (2) a Dual-Task Head Mechanism (DH); and (3) an Efficient Pareto Gradient Optimization (GO) method. Experiments on GPDS-10000 datasets show that FedMH achieves state-of-the-art performance, outperforming relative heterogeneous federated learning baselines.

**Strengths:**

1.	The paper introduces a well-motivated multi-task head design that simultaneously handles user identification and forgery detection—two interrelated but conflicting objectives in signature verification.

2.	FedMH is explicitly designed to support heterogeneous client models. Its gradient optimization protocol allows efficient and stable aggregation across diverse architectures.

3.	The paper provides both rigorous convergence proofs and extensive experiments on benchmark datasets, demonstrating clear superiority over several state-of-the-art baselines.

**Weaknesses:**

1.	The oversampling-based augmentation method, presented as one of the innovations of this paper, lacks sufficient originality. In the ablation study, DA includes both oversampling and other conventional augmentation techniques, making it unclear which specific component contributes to the improvement shown in Tab. 3.

2.	Dual-task learning like this is relatively common in signature verification, and gradient-based dynamic loss weighting has been widely adopted in deep learning tasks. Thus, this contribution is not sufficiently novel. Moreover, the paper lacks an ablation experiment to demonstrate how much advantage the dynamic weighting method provides compared with a static weighting strategy.

3.	In Eq.(8), the definition of the superscript i for g is inconsistent with its earlier usage in the text.

4.	Since the gradient set is recalculated in each round, it would be helpful to present the percentage curve of how often clients select S_sync, S_boundary, or S_bias over training rounds. This visualization would help verify the claimed superiority of the proposed GO scheme.

5.	The paper does not employ recent state-of-the-art models in offline signature verification task. (e.g., HTCSigNet[1], DetailSemNet[2]).

6.	In the ablation study, when DA is included, the effect of DH is not evident and may even be slightly negative (see Tab. 3, rows 3 and 4: the mean improvement with DH is only around 0.0x%, while the variance increases by a similar magnitude). This result fails to convincingly demonstrate the actual contribution of DH to the overall performance of the proposed model.

[1] Zheng L, et al. HTCSigNet: A Hybrid Transformer and Convolution Signature Network for offline signature verification. Pattern Recognition, 2025.
[2] Shih M C, et al. DetailSemNet: Elevating Signature Verification Through Detail-Semantic Integration. European Conference on Computer Vision, 2024.

**Questions:**

Pls see weaknesses

---

> ### Author Response · Authors · 2025-11-17
> **Instructions and Responses for Paper Revision**
>
> Dear Reviewer,
>
> Thank you for reviewing our paper and providing your suggestions. Below, we will address your questions and explain the changes made to improve the paper.
>
> ---
>
> Q1：
>
> The oversampling-based augmentation method, presented as one of the innovations of this paper, lacks sufficient originality. In the ablation study, DA includes both oversampling and other conventional augmentation techniques, making it unclear which specific component contributes to the improvement shown in Tab. 3.
>
> A1：
>
> We conducted ablation experiments on your question, and the results are shown in Figure 4. Using the oversampling method alone does lead to overfitting. However, when combined with traditional enhancement techniques, it can achieve an effect where 1+1>2. This indicates that the oversampling enhancement method can effectively improve the performance of traditional enhancement techniques.
>
> ---
>
> Q2：
> Dual-task learning like this is relatively common in signature verification, and gradient-based dynamic loss weighting has been widely adopted in deep learning tasks. Thus, this contribution is not sufficiently novel. Moreover, the paper lacks an ablation experiment to demonstrate how much advantage the dynamic weighting method provides compared with a static weighting strategy.
>
> A2:
>
> As you mentioned, although multi-task learning is relatively common in signature verification, models such as HTC SigNet still use static weights for training and are not combined with gradient-based dynamic loss weighting. Moreover, existing heterogeneous federated learning methods, as shown in Figure 6, are not suitable for the scenario of federated offline handwritten signature authentication during communication, and negative optimization will occur in the early stage of communication.
>
> Meanwhile, as per your request, we conducted ablation experiments on DH to demonstrate that dynamic weighting methods offer more advantages compared to static weighting strategies when the model overfits in the later stages, as shown in Figure 5(b). The dynamic λ weights of the DH module can provide more flexible weight choices during the commonly used two-headed training for signatures, and further optimize when other static weight settings exhibit overfitting in the later training stages (90-100 epochs).
>
> ---
>
> Q3：
>
> In Eq.(8), the definition of the superscript i for g is inconsistent with its earlier usage in the text.
>
> A3：
>
> Your opinion is very correct. In the content before formula (8), $g_{c}^{(i)}$ is used to represent the gradient of the classification task for the $i$-th client. Here, $i$ is an index ranging from 0 to $n-1$ (where $n$ is the total number of clients), representing a specific client. In formula (8), $G_{c}^{(i)}$ is defined as the cumulative sum of the first $i$ gradients (from $k=0$ to $k=i$). Here, $i$ is an index ranging from 0 to $n-1$, representing the endpoint of the cumulative sum. To avoid misunderstanding, we have changed $i$ and $j$ in formula (8) to $m$.
>
> \begin{equation}
> G_{c}^{(m)} = \sum_{k=0}^{m} g_{c}^{(k)} \quad m \in \{0,1,\dots,n-1\},
> \quad
> G_{v}^{(m)} = \sum_{k=0}^{m} g_{v}^{(k)} \quad m\in \{0,1,\dots,n-1\},
> \end{equation}
>
> ---
>
> Q4：
>
> Since the gradient set is recalculated in each round, it would be helpful to present the percentage curve of how often clients select S_sync, S_boundary, or S_bias over training rounds. This visualization would help verify the claimed superiority of the proposed GO scheme.
>
> A4：
>
> Thank you very much for your suggestion. We have been working hard to find ways to demonstrate the advantages of the GO scheme. Following your suggestion, we have added an analysis experiment on the selected gradient sets. As can be seen from Figure 3, almost all of the optimally selected gradient sets in the training rounds are included in S_sync, S_boundary, or S_bias. It is worth noting that the gradient combinations selected in the early stages of training are often the \(S_{\text{sync}}\) synchronous optimization set, and the gradient combinations selected later are the $S_{\text{boundary}}$ boundary extremum set. This indicates that in the early stages of training, the user classification task and the authenticity identification task can collaborate with each other, while later on, the gradients generated by the authenticity identification task often meet the Pareto verification requirements. This is consistent with the situation where the authenticity signature identification task is dominant in offline signature identification tasks.
>
> ---
>
> Q5：
>
> The paper does not employ recent state-of-the-art models in offline signature verification task. (e.g., HTCSigNet[1], DetailSemNet[2]).
>
> A5：
>
> Now, we are conducting experiments using the most advanced model you provided. However, due to the significant amount of time required, we have only completed part of it so far. We will try our best to finish it during the discussion and add it to the paper. It is expected to be completed by November 20th.
>
> ---

---

> > ### Author Response · Authors · 2025-11-17
> > **add**
> >
> > Q6：
> > In the ablation study, when DA is included, the effect of DH is not evident and may even be slightly negative (see Tab. 3, rows 3 and 4: the mean improvement with DH is only around 0.0x%, while the variance increases by a similar magnitude). This result fails to convincingly demonstrate the actual contribution of DH to the overall performance of the proposed model.
> >
> > A6：
> > Based on your question, we have added an analysis of the ablation experiments. As shown in Table 11, in the heterogeneous scenario, the marginal contributions of the DH and GO modules increase significantly. The DA module alone reduces the EER from $9.96 \pm 0.46$ to $8.05 \pm 0.35$. After adding the DH and GO modules, the EER is further reduced to $7.43 \pm 0.62$, resulting in an additional EER reduction of 0.62%. Compared with the marginal gain of 0.3% in the homogeneous setting (5.58 $\rightarrow$ 5.28), the gain in the heterogeneous setting (0.62%) is more than twice as much.
> >
> > Table 11： The results of the ablation experiments using the HSigNet model and the GPDS-300 dataset, where the performance metrics are the average of the performance metrics of the 5 client models in 5-fold cross-validation. Here, \textbf{DA} refers to the data augmentation module, \textbf{DH} is the dual-task head mechanism, and \textbf{GO} refers to the gradient optimization module.
> > | DA | DH | GO | Performance (EER %) |
> > |----|----|----|----------------------|
> > | ×  | ×  | ×  | \$9.96 \pm 0.46\$    |
> > | √  | ×  | ×  | \$8.05 \pm 0.35\$    |
> > | √  | √  | ×  | \$7.52 \pm 0.55\$    |
> > | √  | √  | √  | \$7.43 \pm 0.62\$    |
> >
> > Your suggestions have helped us address some ambiguities in the proposed FedMH framework. We are very grateful and look forward to your reply.

---

> > ### Author Response · Authors · 2025-11-19
> > **We have increased the performance experiments on HTCSigNet**
> >
> > Dear reviewers,
> >
> > We have added the performance experiments using the HTCSigNet model in Table 1.

---

> > ### Author Response · Authors · 2025-11-21
> > **Supplementary notes for Q1**
> >
> > Dear Reviewer,
> >
> > Regarding the DA module, we have added the results of ablation experiments under different sampling strategies. The results are shown in Figure 4(b). From this, we can see that the sampling strategy is also very important for the data augmentation effect. With the same amount of data after sampling, only the sampling strategy in the DA module achieved better results than directly using data transformation, which indicates that the sampling strategy in DA is also necessary.

---

### Official Review · Reviewer_uqhD · 2025-11-01

**Soundness:** 2
**Presentation:** 3
**Contribution:** 2
**Rating:** 6
**Confidence:** 4

**Summary:**

The paper proposes FedMH, a new federated learning (FL) framework designed for offline handwritten signature verification, which faces data isolation, class imbalance, and model heterogeneity across clients.
FedMH integrates three innovations:
	1.	Client-adaptive data augmentation (DA) — selectively oversamples and transforms minority (forged) samples to balance local datasets.
	2.	Dual-task head mechanism (DH) — models user identification and forgery detection jointly under a Pareto-optimal multi-task formulation, dynamically adjusting loss weights between the two heads.
	3.	Pareto gradient optimization (GO) — aggregates gradients across heterogeneous clients via a Pareto improvement criterion, ensuring stable and efficient global updates without requiring model homogeneity.

The method includes a theoretical convergence proof and extensive experiments on GPDS-10000, Bengali, and Hindi signature datasets. FedMH reportedly achieves state-of-the-art performance in both homogeneous and heterogeneous FL setups, with stronger cross-dataset generalization and lower equal error rate (EER) than baseline methods such as FedProto, FedGH, FedGen, and FedTGP.

**Strengths:**

Originality
	•	The idea of coupling multi-task heads (classification + verification) within federated learning using Pareto-stationary optimization is both novel and meaningful.
	•	The gradient-based Pareto aggregation bridges a gap between multi-objective optimization and heterogeneous FL, offering a creative and technically sound extension of existing frameworks.
	•	The paper reframes signature verification (traditionally single-task) into a dual-task federated optimization problem — a fresh problem formulation that broadens FL’s applicability to biometric domains.

Quality
	The methodology is rigorous and well-supported.
	•	Clear derivations for adaptive λ-weighting between task heads (Eq. 7).
	•	A convergence theorem (Theorem 1) under standard smoothness and bounded gradient assumptions with a formal proof in Appendix C.
	The experimental evaluation is strong:
	•	Covers multiple datasets and both homogeneous and heterogeneous architectures (SigNet, OctConvNet, ViT).
	•	Uses standard signature metrics (AUC, FAR, FRR, EER).
	•	The reproducibility statement and open anonymous code repository further enhance credibility.

Clarity
	•	The mathematical sections are explicit, and appendices provide detailed derivations and algorithms.
	•	The motivation is logically presented: from data imbalance to dual-task conflict to gradient heterogeneity.

Significance
	•	Federated handwritten signature verification is a practical and underexplored domain with strong privacy requirements.
	•	FedMH provides a generalizable paradigm for federated multi-task learning, not limited to signatures — potentially applicable to other privacy-sensitive biometric or multi-label problems.

**Weaknesses:**

Comparative novelty relative to multi-objective FL works:
	•	While Pareto-based optimization is novel in this context, related works such as MOFL (Hu et al., 2022) and FedMOO have explored similar gradient alignment ideas.
	•	The paper could more explicitly position FedMH’s contribution relative to such multi-objective FL methods and justify why its Pareto update is distinct or superior.
	2.	Computational complexity and communication overhead:
	•	The paper does not report time or communication cost, which may be substantial in large-client settings.
	3.	Limited scope of datasets:
	•	Although GPDS, Bengali, and Hindi are standard, they represent a narrow range of handwriting and cultural variability. Testing on other large-scale biometric datasets (e.g., CEDAR, MCYT) would strengthen generalization claims.

	4.	Interpretability of Pareto optimization:
	•	The intuition behind Pareto-front dynamics could be more accessible to non-specialists; Figure 1d and the equations are clear but somewhat dense without geometric interpretation.
	5.	Minor clarity issues:
	•	The conclusion admits the method “needs improvement in convergence speed,” but no empirical evidence of slowdown is shown.

**Questions:**

1. On Pareto gradient optimization efficiency:
	•	How does the computational cost of the Pareto search (Eq. 10–13) scale with client number n?
	•	Could the authors report average per-round latency or communication load compared to FedAvg or FedProto?
	2.	On λ adaptation behavior:
	•	Does λ* converge toward a stable value, oscillate, or correlate with task difficulty?
	•	Could the data augmentation module generalize beyond signature verification to other image-based FL tasks?
	3.	On theoretical assumptions:
	•	The convergence proof assumes β-smoothness and bounded gradients. How realistic are these assumptions for deep CNN/Transformer backbones in practice?
	4.	Reproducibility details:
	•	The paper provides a link to code, but not the license or full dataset preparation scripts. For the camera-ready version, include explicit instructions and hyperparameter settings.

---

> ### Author Response · Authors · 2025-11-17
> **Instructions and Responses for Paper Revision**
>
> Dear Reviewer,
>
> Thank you for reviewing our paper and providing your suggestions. Below, we will address your questions and explain the changes made to improve the paper.
>
> ---
>
> Q1：
>
> On Pareto gradient optimization efficiency:
>
> • How does the computational cost of the Pareto search (Eq. 10–13) scale with client number n?
>
> • Could the authors report average per-round latency or communication load compared to FedAvg or FedProto?
>
> A1:
>
> Your question is very good. We have added relevant experiments as shown in Table 12. The computational overhead of the GO module is highly dynamic and related to two key factors. First, the maximum (searching all S_opt) TFLOPs increases with the number of clients $n$, rising from 96.44 TFLOPs when $n=2$ to 151.63 TFLOPs when $n=5$, which empirically supports our $O(n)$ complexity analysis. Second, the overhead is not fixed in each round, which is clearly reflected in the huge gap between the minimum (searching part of S_opt) (38.34 TFLOPs) and maximum (151.63 TFLOPs) costs when $n=5$. This depends on whether there exists a gradient combination in the set that satisfies the Pareto verification. The communication consumption is shown in Table 8, and computational costs are in Table 9.
>
> **Table 12:** The TFLOPs results of different clients of the GO module under settings with different numbers of clients. The calculation cost is related to the number of clients $n$ and the Pareto index $(i, j)$ dynamically selected during the search process.
>
> | Clients (n) | Number of Searches  | Reported TFLOPs |
> |-------------|-----------------------|------------------|
> | \$n=2\$     | min                   | 38.58 TFLOPs    |
> | \$n=2\$     | max                   | 96.44 TFLOPs    |
> | \$n=4\$     | min                   | 136.47 TFLOPs   |
> | \$n=4\$     | max                   | 136.47 TFLOPs   |
> | \$n=5\$     | min                   | 38.34 TFLOPs    |
> | \$n=5\$     | max                   | 151.63 TFLOPs   |
>
> ---
>
> Q2：
>
> • Does λ* converge toward a stable value, oscillate, or correlate with task difficulty?
>
> • Could the data augmentation module generalize beyond signature verification to other image-based FL tasks?
>
> • On theoretical assumptions: The convergence proof assumes β-smoothness and bounded gradients. How realistic are these assumptions for deep CNN/Transformer backbones in practice?
>
> • Reproducibility details:  The paper provides a link to code, but not the license or full dataset preparation scripts. For the camera-ready version, include explicit instructions and hyperparameter settings.
>
> A2:
>
> Your question has helped us further explore the role of the DH module. From Figure 5 (c), it can be found that λ* converges from around 0.98 to around 0.84 during the training phase, and from Figure 5 (b), we can see that the change of λ* indeed provides the effect of overfitting suppression for the later stage of the model.
>
> For classification tasks, the DA module can also play a similar role. This is because the DA module is essentially an automatic class balancing strategy, which has a certain inhibitory effect on the label drift phenomenon in federated learning.
>
> Regarding the bounded gradient, this assumption is generally considered relatively reasonable in the practice of major models. Although gradients may theoretically explode, in actual training, this constraint can be easily enforced through techniques such as gradient clipping. In stable training, gradients usually remain within a reasonable range. However, concerning β-smoothness, the global smoothness constant of major models may not exist or could be very large. Nevertheless, theoretical analyses typically aim to prove the behavior of algorithms in local regions (the regions where the optimization trajectory lies). Therefore, we have no choice but to use the β-smoothness assumption commonly employed in previous federated learning methods to analyze the local convergence[1].
>
> [1] Yun-Hin Chan, Rui Zhou, Running Zhao, Zhihan Jiang, and Edith C-H Ngai. Internal cross-layer gradients for extending homogeneity to heterogeneity in federated learning. In Proceedings of International Conference on Learning Representation, pp. 1–29, 2024.
>
> We have re-updated the anonymous repository and provided a detailed README document with parameter explanations.
>
> ---
>
> Your in-depth research and suggestions on the theory have helped us improve our understanding of the proposed FedMH method. Thank you again for your suggestions.

---

> > ### Author Response · Authors · 2025-11-18
> > **We will reply to the Weaknesses you have pointed out as soon as possible.**
> >
> > Dear Reviewer,
> >
> > We are actively preparing relevant work in response to the Weaknesses you pointed out and will reply as soon as possible.

---

> ### Author Response · Authors · 2025-11-19
> **We have completed the work related to Weaknesses and are replying here.**
>
> ---
> W1:
>
> Comparative novelty relative to multi-objective FL works:
>
> • While Pareto-based optimization is novel in this context, related works such as MOFL (Hu et al., 2022) and FedMOO have explored similar gradient alignment ideas.
>
> • The paper could more explicitly position FedMH’s contribution relative to such multi-objective FL methods and justify why its Pareto update is distinct or superior.
>
> A1:
>
> We have added Appendix F in the appendix, and in Appendix F1, we further explain and discuss the differences and advantages of the proposed FedMH method compared with previous multi-objective federated learning methods through textual descriptions and Figures 10 and 11.
>
> ---
>
> W2:
>
>  Computational complexity and communication overhead:
>     • The paper does not report time or communication cost, which may be substantial in large-client settings.
>
> A2:
>
> We have added Tables 8 and 9 to illustrate the reporting time or communication cost.
>
> ---
>
> W3:
>
>  Limited scope of datasets:
>     • Although GPDS, Bengali, and Hindi are standard, they represent a narrow range of handwriting and cultural variability. Testing on other large-scale biometric datasets (e.g., CEDAR, MCYT) would strengthen generalization claims.
>
> A3:
>
> We have added experiments with relevant datasets in Figure 2(d).
>
> ---
>
>
> W4:
>
>  Interpretability of Pareto optimization:
>     • The intuition behind Pareto-front dynamics could be more accessible to non-specialists; Figure 1d and the equations are clear but somewhat dense without geometric interpretation.
>
> A4:
>
> We have added Appendix F.2 and used Figures 10 and 11 to illustrate why Pareto update is used, as well as explain Pareto update from the perspective of parameter update.
>
> ---
>
> W5:
>
>  Minor clarity issues:
>     • The conclusion admits the method “needs improvement in convergence speed,” but no empirical evidence of slowdown is shown.
>
> A5:
>
> We have revised the description in the conclusion. As can be seen in Figure 6, the proposed FedMH method converges more slowly compared to other methods. This is due to the dynamic nature of the DH and GO components. However, from ablation experiments such as those in Figure 5, it can be observed that this plays a positive role in further improving the model performance.
>
> ---

---

### Official Review · Reviewer_vypZ · 2025-11-01

**Soundness:** 3
**Presentation:** 3
**Contribution:** 3
**Rating:** 6
**Confidence:** 4

**Summary:**

The authors proposes federated learning based on a multi-task head strategy (FedMH) method for offline handwritten signature verification under limited data, class imbalance, and heterogeneous client models. It has 3 main components: (1) an adaptive data augmentation module to address intra-client class imbalance by targeting few-shot classes in local single-user genuine and forged signature subsets;  (2) a dual-task head mechanism that jointly optimizes user classification and forgery detection; and  (3) an efficient Pareto gradient optimization protocol employing linear probing in the parameter space for efficient knowledge aggregation across heterogeneous models. The proposed FedMH method is validated on GPDS, Bengali, and Hindi signature verification datasets, and results show it can outperforms SOTA heterogeneous FL baselines, e.g., FedGH, FedTGP, FedProto, in both homogeneous and heterogeneous setups.

**Strengths:**

+ Overall, the paper is clearly written, well organized, and easy to follow. FedMH formulates federated offline signature verification as an iterative process with Pareto-stationary solutions for multi-task heads.  The dual-task head is employed to update the model on local clients. It aligns task-specific gradients, and guides the model toward a Pareto stationary solution that balances the performance of both tasks.
+ The application and challenges of federated learning in offline handwritten signature verification is well justified by the authors.
+ A convergence proof is provided, showing that FedMH converges to a Pareto-stationary point under standard assumptions. This ensures the reliability and stability of FedMH.
+ Experiments on GPDS, Bengali, and Hindi datasets validate the effectiveness of FedMH. It achieves SOTA performance compared to heterogeneous FL baseline methods. When faced with unfamiliar datasets, FedMH also demonstrates better performance than baseline methods. The ablation studies show the benefits from each module, although the client-adaptive minority class data augmentation (to balance/enhance local data) provides the largest improvement.
+ The Appendix has additional information on FedMH properties, experimental settings., hyper-parameter settings, ablation studies, analysis of computational complexity, feature visualizations, description of CASE components, analysis of novelty, and experimental results that help support the paper.

**Weaknesses:**

- The methodology (Section 3) is dense and difficult to follow. Although FedMH represents an effective method to addresses heterogeneity and multi-task coupling in offline signature verification, it extends on existing methods (augmentation, Pareto optimization, dual-head training).
- The experimental validation raises some concerns. The results in Table 1 and 2 should be replicated multiple times using some cross-validation procedures.  The authors should consider using performance measures based on the precision-recall curve (as opposed to the ROC curve) to observe the impact on performance of class imbalance.
- The choice of some loss terms and algorithmic components (e.g., thresholds and batch size) appear heuristic.  Their sensitivity should be analyzed empirically with more ablations studies.
- Limited discussion and interpretation of experimental results in Section 4.  This paper should also contain an experimental analysis of time and memory complexity.  Computational overhead appears higher than SOTA methods yet is not analyzed, raising concerns about scalability.
- Their code is not made available, so there is a concern that the results in this paper would be difficult for a reader to reproduce.

**Questions:**

See my comments in weaknesses.

**Details Of Ethics Concerns:**

None.

---

> ### Author Response · Authors · 2025-11-17
> **Instructions and Responses for Paper Revision**
>
> Dear Reviewer,
>
> Thank you for reviewing our paper and providing your suggestions. Below, we will address your questions and explain the changes made to improve the paper.
>
> ---
>
> Q1：
> The methodology (Section 3) is dense and difficult to follow. Although FedMH represents an effective method to addresses heterogeneity and multi-task coupling in offline signature verification, it extends on existing methods (augmentation, Pareto optimization, dual-head training).
>
> A1:
> The FedMH framework is indeed built upon existing methods such as data augmentation, Pareto optimization, and dual-head training. However, as can be seen from the additional ablation experiments in Figure 5, Figure 4, etc., the FedMH method has effectively improved these methods and integrated them into the federated offline handwritten signature verification environment.
>
> ---
>
> Q2:
> The experimental validation raises some concerns. The results in Table 1 and 2 should be replicated multiple times using some cross-validation procedures. The authors should consider using performance measures based on the precision-recall curve (as opposed to the ROC curve) to observe the impact on performance of class imbalance.
>
> A2:
> Your opinion is very reasonable, but the personalized experiments (Tables 1 and 2) are designed to simulate a specific and realistic Federated Learning (FL) scenario, where data is distributed among 5 clients in a non-independent and identically distributed (non-IID) manner, achieved through the Dirichlet(0.1) distribution.
> Performing k-fold cross-validation by pooling all data from 700 users, re-dividing them into k parts, and then re-distributing them to 5 clients using the Dirichlet distribution would not evaluate the model's performance in a fixed federated environment, but rather the average performance across k different federated environments.  Additionally, re-dividing only the data within each client would also affect the original user distribution.
>
> Considering this, we had no choice but to follow the standard practices of personalized FL benchmarking in the experiments for Tables 1 and 2. For each user's data, we divided it according to a fixed ratio and took the average of the results from the last 10% of global rounds. This ensures that the test set is strictly unseen and is tailored to the same user distribution. Furthermore, we compensated by using multiple cross-validation in the generalization experiments (Figure 2) with other datasets.
>
> The metrics we currently use (AUC, EER, FAR, FRR) are standard evaluation metrics in the field of signature verification (for example, HTCSigNet[1]). In signature authentication, FAR refers to the situation where the system mistakenly accepts forged signatures, which may threaten security. FRR refers to the situation where the system mistakenly rejects genuine signatures, which may affect user convenience. EER is the threshold point where FRR equals FAR, representing the balance between the system's security and convenience. In contrast, although precision and recall are more commonly used, they do not directly correspond to the decision scenarios in signature applications.
>
> ---
>
> Q3:
> The choice of some loss terms and algorithmic components (e.g., thresholds and batch size) appear heuristic. Their sensitivity should be analyzed empirically with more ablations studies.
>
> A3:
> Your opinion is very correct. We have added ablation experiments on hyperparameters and experimental analysis of components. The results are shown in Figure 4, Figure 5, and Figure 8.
>
> ---
>
> Q4:
> Limited discussion and interpretation of experimental results in Section 4. This paper should also contain an experimental analysis of time and memory complexity. Computational overhead appears higher than SOTA methods yet is not analyzed, raising concerns about scalability.
>
> A4:
> Your concern is very reasonable. FedMH does incur higher resource overhead, as shown in Tables 8 and 9. This is because, to address the unique dual-task challenge in signature verification, the FedMH method transmits dual-head gradients simultaneously and performs dynamic linear probe searches during this process. Given that the FedMH method achieves SOTA-level EER and effectively suppresses negative transfer, it is worthwhile to exchange controllable computational resources for significant improvements in security and accuracy. Of course, we have also identified this as a key direction for future improvements to this work.
>
> ---
>
> Q5:
> Their code is not made available...
>
> A5:
> We have reorganized the code into the anonymous repository link and placed the weight link in the supplementary file.
>
> ---
>
> [1] Zheng L, et al. HTCSigNet: A Hybrid Transformer and Convolution Signature Network for offline signature verification. Pattern Recognition, 2025.
>
> Thank you again for your careful review of our paper and your profound insights. We look forward to receiving more of your suggestions.

---

> > ### Author Response · Authors · 2025-11-17
> > **Supplementary information for Q2**
> >
> > In response to your suggestion, "The authors should consider using performance measures based on the precision-recall curve (as opposed to the ROC curve) to observe the impact on performance of class imbalance," we will add experiments on the precision-recall curve, around November 20th. Your suggestion has helped improve our work, and we look forward to further communication with you.

---

> > > ### Author Response · Authors · 2025-11-19
> > > **We have added the precision-recall curve for question 2.**
> > >
> > > Dear Reviewer,
> > >
> > > We have added the precision-recall curve (Figure 9). The experimental results show that the proposed FedMH method in this paper achieves rapid convergence in the early stage of training and maintains significantly better performance stability than the comparison baselines throughout the communication process. It overcomes the performance fluctuations and negative optimization phenomena that occur in other heterogeneous federated learning methods, such as LG-FedAvg and FedProto, in the middle and late stages of training due to gradient conflicts or catastrophic forgetting caused by model heterogeneity. It is worth noting that the precision and recall curves of FedMH show a high degree of consistency, which is attributed to the experiment following the dynamic threshold selection strategy based on the Equal Error Rate (EER) criterion in the scenario of handwritten signature verification [1]. That is, on an approximately balanced test set containing 10 genuine signatures and 12 forged signatures, the False Acceptance Rate (FAR) and False Rejection Rate (FRR) are forced to be balanced.
> > >
> > > [1] Luiz G. Hafemann, Robert Sabourin, and Luiz S. Oliveira. Learning features for offline handwritten signature verification using deep convolutional neural networks. Pattern Recognition, 70:163–176, 2017.

---

### Author Response · Authors · 2025-11-13
**We will improve our work according to the comments as soon as possible and reply.**

Dear reviewers and area chairs,

Thank you for your professional and insightful comments, which have benefited us a great deal. We will improve the content of the paper as soon as possible in accordance with your suggestions and answer your questions one by one.

---

### Author Response · Authors · 2025-11-17
**We have initially completed the revision of the article and replied**

Dear reviewers and area chairs,

We have initially completed the revision of the paper. Thank you again for your professional and insightful suggestions, which have helped us further improve our work. We are ready to answer any questions you may have at any time.

---

### Author Response · Authors · 2025-11-25
**Summary of Thesis Revisions**

Dear reviewers and area chairs,

Regarding the ablation questions raised by the reviewers, we hereby provide a summary response.

---
Q1: In Table 3, the effect of combining the DH and GO components with the DA component is not obvious.

Based on your question, we have added an analysis of the ablation experiments. As shown in Table 11, in the heterogeneous scenario, the marginal contributions of the DH and GO modules increase significantly. The DA module alone reduces the EER from $9.96 \pm 0.46$ to $8.05 \pm 0.35$. After adding the DH and GO modules, the EER is further reduced to $7.43 \pm 0.62$, resulting in an additional EER reduction of 0.62%. Compared with the marginal gain of 0.3% in the homogeneous setting (5.58 $\rightarrow$ 5.28), the gain in the heterogeneous setting (0.62%) is more than twice as much.

This finding confirms that although DA is used to handle the problem of data imbalance, the DH and GO modules are crucial for promoting effective knowledge aggregation among heterogeneous clients, enabling the global model to better utilize the gradients from high-performance clients to assist weaker clients.

Moreover, **compared with other baseline methods that use feature prototypes for aggregation**, the proposed FedMH method indeed **improves model performance through communication between clients and the server**, rather than having a negative effect.

**Table 11:** The results of the ablation experiments using the HSigNet model and the GPDS-300 dataset, where the performance metrics are the average of the performance metrics of the 5 client models in 5-fold cross-validation. Here, **DA** refers to the data augmentation module, **DH** is the dual-task head mechanism, and **GO** refers to the gradient optimization module.

| DA | DH | GO | Performance (EER %) |
|----|----|----|----------------------|
| ×  | ×  | ×  | \$9.96 \pm 0.46\$    |
| √  | ×  | ×  | \$8.05 \pm 0.35\$    |
| √  | √  | ×  | \$7.52 \pm 0.55\$    |
| √  | √  | √  | \$7.43 \pm 0.62\$    |


Meanwhile, we also added the DA component to the related methods that use prototype aggregation, **because such methods achieved the worst results in Table 2**. According to the results shown in Table 13, although the DA module has a positive effect on these methods, their performance after using the DA method is still weaker than that of other methods in Table 2. This indicates that there is an upper limit to the performance enhancement effect of the DA module, and a good communication and training strategy (GO module, DH module) is necessary.

**Table 13:** The enhancement effect of the DA module on federated learning methods under the HSigNet model settings and the GPDS-700 dataset.

| Method   | Use DA       |          |          |          | Not Use DA   |          |          |          |
|:-------:|:-----------:|:--------:|:--------:|:--------:|:-----------:|:--------:|:--------:|:--------:|
|          | EER          | AUC      | FAR      | FRR      | EER          | AUC      | FAR      | FRR      |
| FedProto | 9.44         | 93.67    | 9.64     | 9.24     | 13.6         | 90.8     | 13.9     | 13.3     |
| FedKD    | 13.18        | 90.18    | 13.74    | 12.62    | 18.4         | 85.8     | 19.2     | 17.6     |
| FedTGP   | 9.04         | 94.53    | 9.23     | 8.84     | 16.1         | 88.3     | 16.4     | 15.7     |


---

Q2: Ablation of the DA component

For the adaptive data augmentation (DA) module, by comparing different augmentation configurations and sampling strategies, we confirmed its necessity as a fundamental means to address local data imbalance. As shown in Figure 4(a), the complete strategy (ALL) that combines oversampling and data transformation achieved the lowest and most stable equal error rate (EER), while using only oversampling would lead to severe overfitting and performance degradation.

Meanwhile, as shown in Figure 4(b), the "Balance" oversampling strategy for minority classes used in the DA module achieved the best effect. This indicates that data transformation in the DA module is as important as the oversampling strategy for minority classes, and both play a role.

---

Q3: Ablation of the DH component

For the DH module, we verified through experiments its key role in coordinating user classification and authenticity verification tasks and preventing overfitting in the later stages of training. As shown in Figure 5(a), the parallel dual-head training mode with dynamic weights outperforms the single-task head and sequential training modes throughout the entire process, especially effectively avoiding the performance degradation of the sequential mode after 70 epochs due to gradient conflicts.

In addition, as shown in Figure 5(b), compared with the static weight setting, the strategy of dynamically adjusting the task weight $\lambda$ endows the model with the ability to flexibly balance tasks in the later stages of training (90-100 epochs), thereby achieving a more robust convergence effect.

---

---

### Note · Program_Chairs · 2026-01-17
**Submission Desk Rejected by Program Chairs**

The following references in this submission do not refer to real documents and/or have major errors in bibliographic information:

 Hongyi Chang, Shaoxiang Wang, Jakub Konečný, Sen Yu, Gauri Joshi, and Virginia Smith. Communication-efficient federated learning via knowledge distillation. IEEE Journal of Selected Topics in Signal Processing, 14(6):1183-1194, 2020.